# Prefusion-stabilized SARS-CoV-2 S2-only antigen provides protection against SARS-CoV-2 challenge

Ching-Lin Hsieh [1,7], Sarah R. Leist [2,7], Emily Happy Miller [3,4], Ling Zhou[1], John M. Powers [2], Alexandra L. Tse [3], Albert Wang[3], Ande West[2], Mark R. Zweigart[2], Jonathan C. Schisler [5], Rohit K. Jangra[3,6], Kartik Chandran[3], Ralph S. Baric [2] & Jason S. McLellan [1] ✉

Ever-evolving SARS-CoV-2 variants of concern (VOCs) have diminished the effectiveness of therapeutic antibodies and vaccines. Developing a coronavirus vaccine that offers a greater breadth of protection against current and future VOCs would eliminate the need to reformulate COVID-19 vaccines. Here, we rationally engineer the sequence-conserved S2 subunit of the SARS-CoV-2 spike protein and characterize the resulting S2-only antigens. Structural studies demonstrate that the introduction of interprotomer disulfide bonds can lock S2 in prefusion trimers, although the apex samples a continuum of conformations between open and closed states. Immunization with prefusion-stabilized S2 constructs elicits broadly neutralizing responses against several sarbecoviruses and protects female BALB/c mice from mouse-adapted SARS-CoV-2 lethal challenge and partially protects female BALB/c mice from mouse-adapted SARS-CoV lethal challenge. These engineering and immunogenicity results should inform the development of next-generation pan-coronavirus therapeutics and vaccines.

The COVID-19 pandemic has led to more than 760,000,000 cases and 6,900,000 deaths globally since the severe acute respiratory syndrome coronavirus 2 (SARS-CoV-2) first emerged in the human population in late 2019. SARS-CoV-2 is continuously evolving to improve fitness and escape existing immunity elicited by vaccinations or prior natural infections[1]. Polyclonal antibodies from vaccinated or infected individuals, as well as FDA-approved therapeutic monoclonal antibodies, have become less effective at providing protection from earlier and contemporary Omicron variants XBB.1.5 and BQ.1.1[2,3]. Even vaccinations with second-generation boosters based on BA.1 or BA.4/5 are not sufficient to elicit robust neutralizing antibody responses to

the XBB.1.5 and BQ.1.1 variants[4–7]. Therefore, novel vaccine platforms and strategies that can provide greater breadth against SARS-CoV-2 VOCs, and human betacoronaviruses in general, are urgently needed.

As the primary target of antibody-mediated neutralization, the spike (S) protein from SARS-CoV-2 is the main antigen in approved vaccines[8–10]. The S protein is composed of the S1 subunit, which mediates attachment, and the S2 subunit, which fuses the viral and host cell membranes. The S1 subunit contains an N-terminal domain (NTD), a receptor-binding domain (RBD), and two subdomains (SD1 and SD2)[11,12]. Neutralizing antibodies from human sera mainly target the RBD and NTD[13,14], resulting in immune pressure and selection that

[1]Department of Molecular Biosciences, The University of Texas at Austin, Austin, TX 78712, USA. [2]Department of Epidemiology, University of North Carolina at Chapel Hill, Chapel Hill, NC 27599, USA. [3]Department of Microbiology and Immunology, Albert Einstein College of Medicine, Bronx, NY 10461, USA. [4]Department of Medicine-Infectious Diseases, Albert Einstein College of Medicine, Bronx, NY 10461, USA. [5]McAllister Heart Institute and Department of Pharmacology, The University of North Carolina at Chapel Hill, Chapel Hill, NC 27599, USA. [6]Present address: Department of Microbiology and Immunology, Louisiana State University Health Sciences Center-Shreveport, Shreveport, LA 71103, USA. [7]These authors contributed equally: Ching-Lin Hsieh, Sarah R. Leist. ✉e-mail: jmclellan@austin.utexas.edu

leads to an accumulation of substitutions in the S1 subunit in evolving variants[15]. Antibodies targeting the conserved S2 subunit can be protective and often cross-reactive with spike proteins from diverse betacoronaviruses[16]. For instance, antibodies that bind to the stem helix at the base of the S2 subunit provide cross-protection against all SARS-CoV-2 VOCs[17,18], and some of these antibodies can also neutralize Middle East respiratory syndrome coronavirus (MERS-CoV)[19–21]. Moreover, non-neutralizing antibodies that target S2 elicit protective FcR effector functions that protect from severe disease across distant sarbecoviruses and VOCs[22]. Recently, antibodies that bind the conserved fusion peptide have been isolated and characterized[23–25], and several of these show remarkable breadth. The identification of these antibodies with distinct correlates of protection, combined with the high sequence conservation of the S2 subunit, makes an S2-only antigen an attractive candidate for vaccine development[26].

Similar to influenza hemagglutinin (HA), the fusogenic S2 subunit is metastable and will refold in the absence of the fusion-suppressive S1 domains as it transitions to an elongated, stable postfusion conformation. A variety of strategies, including the introduction of disulfide bonds, have been used to stabilize the HA stem (analogous to S2) in the prefusion conformation[27–29]. Here, we used a highly stable S protein containing six proline substitutions in the S2 subunit (HexaPro)[30] as our prototype to guide structure-based antigen design. Not only do we report the successful stabilization of trimeric, prefusion S2-only proteins and characterize their structure, antigenicity, and immunogenicity, but importantly, demonstrate protection to various extents across antigenically distinct sarbecoviruses in lethal mouse models of human disease.

## Results

### Structure-based design of prefusion-stabilized SARS-CoV-2 S2 subunits

We first generated a base construct for S2-only antigens (HexaPro-S2) by removing the entire S1 subunit from the HexaPro S protein (containing 6 stabilizing proline substitutions) and deleting a flexible region (residues 686–696) from the N-terminus of the S2 subunit. Given that the S1 subunit forms an extensive trimerization interface and belt-like structure that confines the membrane-distal apex of S2, we designed 8 interprotomer disulfide substitutions using the prefusion structure of HexaPro S (PDB ID: 6XKL)[30] as a guide (Fig. 1a). Based on the Cβ distances of residues at the interprotomer interface, the following three sets of substitutions were assessed in the base construct: (1) Y707C/P792C, S704C/K790C, A713C/L894C, and G891C/P1069C near the lateral face, (2) Q755C/N969C and G757C/S968C near the apex, and (3) S1030C/D1041C and G1035C/V1040C in the core of S2. The Y707C/T883C substitution from Olmedillas et al.[31] was included as a comparison. Each construct was characterized for expression yield and interprotomer disulfide formation as assessed by non-reducing SDS-PAGE, monodispersity by size-exclusion chromatography (SEC) and thermostability by differential scanning fluorimetry (DSF).

Four interprotomer disulfide bond substitutions had comparable protein expression relative to HexaPro S2, and one design—G757C/S968C—had a substantial decrease in protein yield (Fig. 1b). All five substitutions formed interprotomer disulfide bonds, but to different extents. Both Y707C/T883C and A713C/L894C showed detectable monomeric and dimeric fractions on the non-reducing gel. Notably, Y707C/P792C and S704C/K790C had 83% and 85% in the trimeric fractions, respectively. In contrast, the expression of Q755C/N969C, S1030C/D1041C, and G1035C/V1040C was completely abolished. Except for Y707C/P792C, all expressed constructs had a slight rightward shift of the SEC peak relative to the base construct, consistent with a more compact conformation (Fig. 1c). Both Y707C/T883C and Y707C/P792C exhibited a broader and less monodisperse peak than others, indicating heterogeneity, which was congruent with the DSF

analysis that revealed multiple melting temperatures (Tm) (Fig. 1d). The substitutions S704C/K790C, A713C/L894C, and G757C/S968C showed substantial increases in Tm relative to the base construct (Fig. 1d and Supplementary Table 1), ranging from +7.1 to +13.7 °C. Next, we added A713C/L894C, G757C/S968C, or both disulfide substitutions on the background of S704C/K790C. The expression of the combinatorial disulfide constructs containing G757C/S968C was completely abolished. Although A713C/L894C/S704C/K790C showed comparable expression relative to S704C/K790C, it exhibited incomplete trimer formation on the non-reducing SDS-PAGE gel and was not pursued further (Supplementary Fig. 1a, b).

Based on the extent of trimer formation, expression yield, and thermostability, we prioritized S704C/K790C for further engineering through adding proline or interprotomer salt-bridge designs. Although the addition of Q895P decreased the expression relative to the parental construct, the Q957E substitution enhanced the expression with the majority of protomers covalently linked via the disulfide bond (Supplementary Fig. 1c, d). Given that several broadly protective antibodies have been demonstrated to target the helical stalk at the base of the globular S2 ectodomain[16–21,32], we hypothesized that transplanting multiple stalks (corresponding to SARS-CoV-2 spike residues 1142–1208) from related betacoronavirus spikes to this construct could have beneficial effects on immunogenicity (note that SARS-CoV and SARS-CoV-2 spikes have identical amino acid sequences in this region). The expression of the TriStalks construct (SARS-CoV-2, MERS-CoV, and HKU1) was lower and the SEC traces were more polydisperse than its parental S704C/K790C construct (Supplementary Fig. 1c, d). In contrast, the expression of PentaStalks (SARS-CoV-2, MERS-CoV, HKU1, OC43, and HKU9) was substantially higher than its parental construct. We further constructed a version of the S704C/K790C S2 antigen without the stem helix and remaining ectodomain stalk (residues 1142–1208), named Δstalk, which also expressed robustly and eluted as a monodisperse peak on SEC (Supplementary Fig. 1c, d). Large-scale expression of our best HexaPro-S2 variant that contains S704C/K790C/Q957E substitutions produced 2.5 mg of protein from 1 L of FreeStyle 293-F cells (Supplementary Fig. 2a). We renamed it HexaPro-SS (stabilized stem) and added the same modifications at the C-terminus of S2, generating HexaPro-SS-PentaStalks and HexaPro-SS-Δstalk. HexaPro-SS-PentaStalks and HexaPro-SS-Δstalk also expressed robustly in a large-scale format (Supplementary Fig. 2b, c). To confirm that our engineering did not alter the antigenic surface of S2, we examined the binding of various S2-specific antibodies to the antigens (Supplementary Fig. 3). As expected, RBD-targeting antibody N3-1[33] did not bind to HexaPro-SS or HexaPro-SS-Δstalk. Previous studies have reported that stem-helix-specific S2 antibodies show broadly neutralizing capability. Importantly, stem-helix antibodies IgG22[21], S2P6[18] and CC40.8[20] bound to HexaPro-SS at similar magnitudes as those observed for binding to the full S ectodomain S-2P and HexaPro, but not HexaPro-SS-Δstalk, which lacks the epitope. Interestingly, fusion-peptide-directed antibodies CoV44-79[23] and CoV91-27 bound strongly to HexaPro-SS and weakly to the full-length spikes (S-2P and HexaPro), suggesting that the fusion peptide is more accessible in the context of S2-only antigens. Collectively, we stabilized SARS-CoV-2 S2-only antigens through structure-based design and expanded the variety of S2-only antigens through additional modifications at the C-terminus to probe their influence on immunogenicity.

### Prefusion-stabilized S2 structures reveal a range of flexibility at the trimer apex

To investigate whether our design effectively locked the S2-only antigen in the prefusion conformation, we determined the crystal structure of HexaPro-SS-Δstalk. The protein complex crystallized in space group R3 and an X-ray diffraction dataset was collected to a resolution of 3.2 Å. One protomer of the S2 subunit from the HexaPro spike

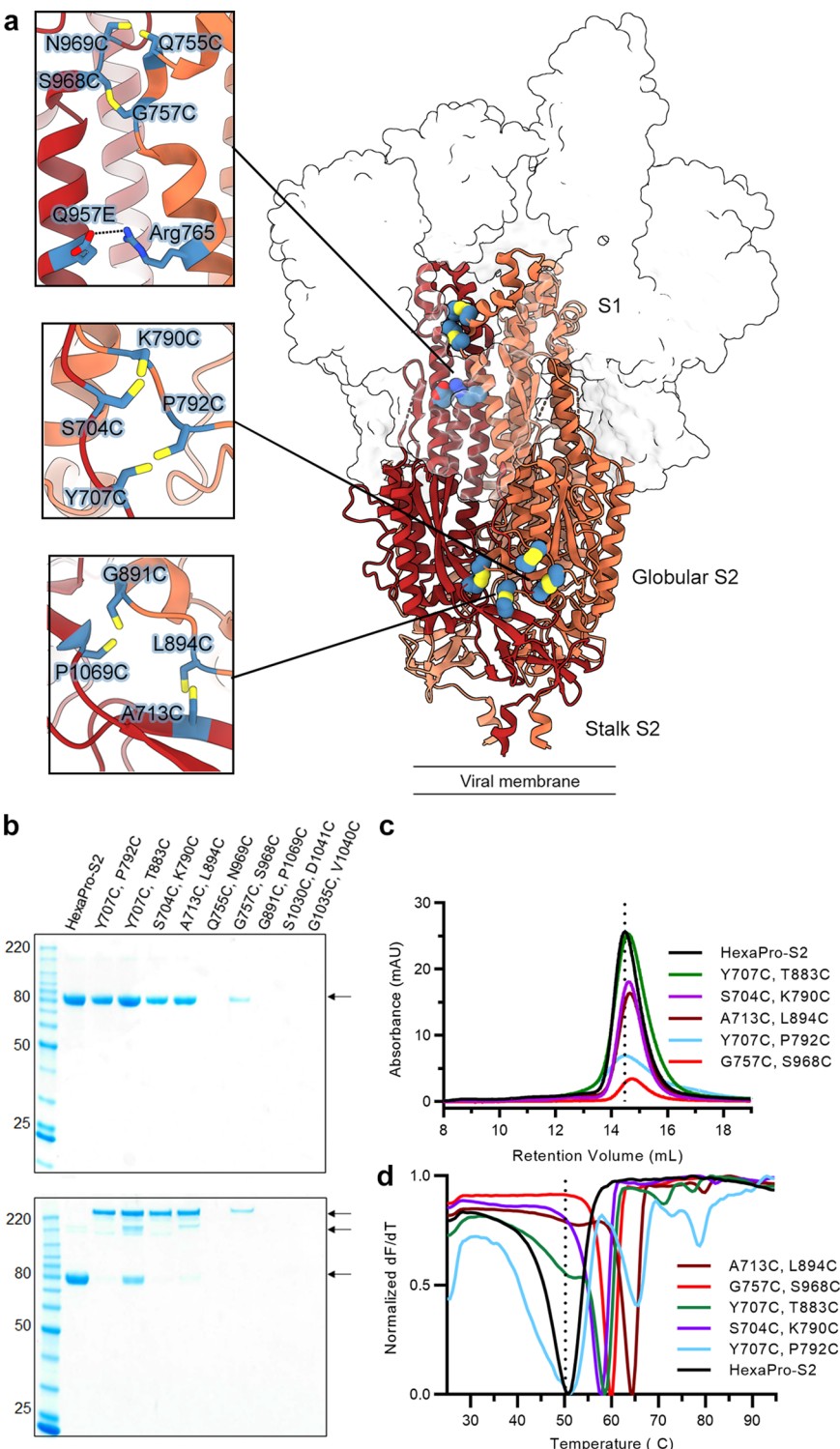

**Fig. 1 | Interprotomer disulfide substitution stabilizes S2-only constructs in a covalently linked trimer. a** Side view of HexaPro (PDB ID: 6XKL). The S1 subunits are shown as a transparent molecular surface. The S2 subunit of each protomer is shown as a ribbon diagram. Each inset corresponds to a zoomed view of interprotomer disulfide designs. Side chains in each inset are shown as sticks with sulfur atoms in yellow. **b** Reducing (top) and non-reducing (bottom) SDS-PAGE analysis of each interprotomer disulfide variant. The molecular weight standards in kDa are indicated at the left. The position of monomer, dimer and trimer bands are indicated at the right. The SDS-PAGE analysis was performed once. **c** Size-exclusion chromatography and **d**, differential scanning fluorimetry analysis of each interprotomer disulfide variant. The vertical dotted line indicates (**c**) the peak retention volume and (**d**) the melting temperature for the HexaPro-S2 containing no disulfide substitution. Source data are provided as a Source Data file.

structure (PDB ID: 6XKL) was used as a search model for molecular replacement. Iterative model building and refinement resulted in a structure with $R_{work}$ and $R_{free}$ values of 22.6% and 26.2%, respectively (Supplementary Table 2). The asymmetric unit contained two S2 protomers (each from a different trimer), packed in a head-to-head arrangement, resulting in the upper half of the central helices packing against each other. HexaPro-SS-Δstalk adopted a prefusion conformation and formed a trimer when crystallographic symmetry was

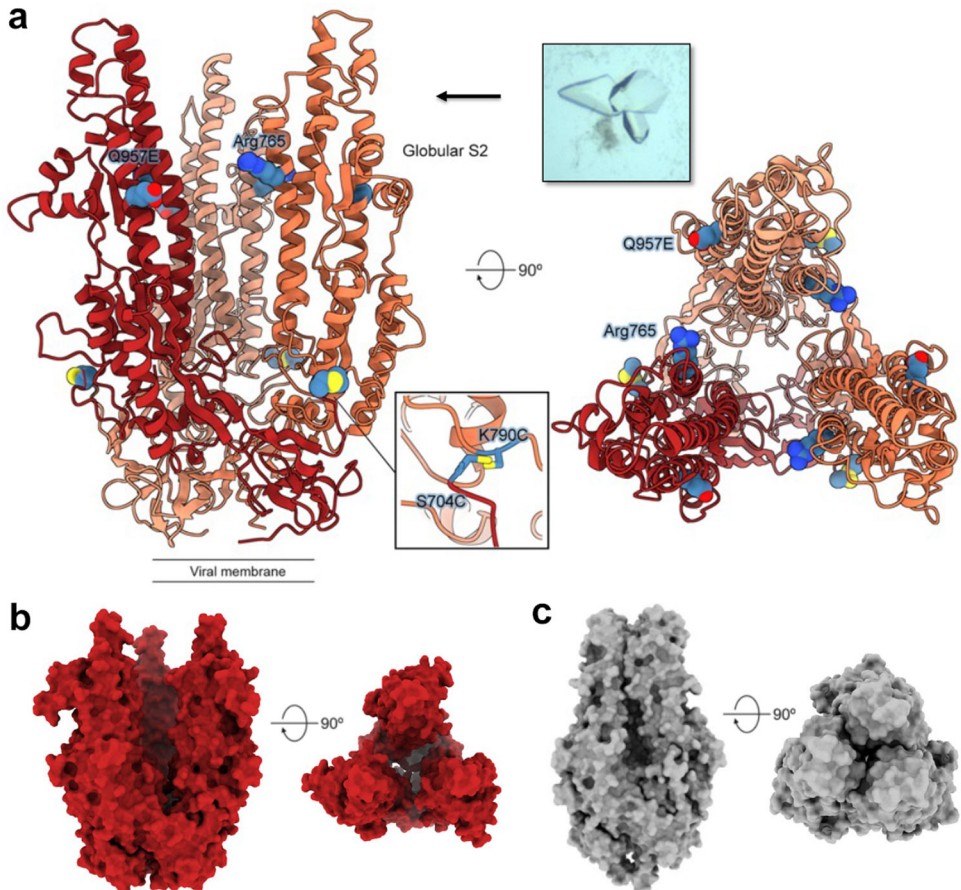

**Fig. 2 | Disulfide-stabilized S2 crystallized in an open prefusion conformation. a** Side (left) and top (right) views of the crystal structure for HexaPro-SS-Δstalk are shown in ribbons. The cysteine substitutions (inset) from symmetry mates are in proximity to each other to form disulfide bonds, but Glu957 does not form a salt bridge with Arg765 from the neighboring protomer. **b** The side and top view of HexaPro-SS-Δstalk presents a prefusion conformation with the trimer apex splayed open. **c** The surface representation of HexaPro (PDB ID: 6XKL) with the S1 subunit removed, exhibiting a closed prefusion conformation.

applied (Fig. 2a). The cysteine substitutions (Cys704/Cys790) from symmetry mates are in proximity to each other to form disulfide bonds, but Glu957 could not form a salt bridge with Arg765 from the neighboring protomer due to separation of the protomers at the apex. In comparison with the S2 subunit from the HexaPro spike structure (PDB ID: 6XKL), HexaPro-SS-Δstalk is splayed apart at the trimer apex with the central helices ~15° further away from the 3-fold axis (Fig. 2b, c). Given that this partially open conformation could result from a crystal packing artifact, we set out to examine whether HexaPro-SS-Δstalk could sample multiple conformations as assessed by cryo-EM. After data collection and processing, we were able to obtain at least three distinct 3D reconstructions, with the apex of S2 adopting closed, semi-open and fully open conformations (Fig. 3). Although the low resolution of these 3D reconstructions prevented us from building atomic models, all three conformations are prefusion trimers, consistent with our crystallographic data. Taken together, the HexaPro-SS-Δstalk S2-only antigen presents multiple prefusion conformations that differ in the extent to which the apex is open.

## Sera from S2-immunized mice neutralize diverse sarbecovirus rVSV-CoVs

To investigate whether our stabilized S2-only antigens could elicit broadly reactive antibody responses, we immunized mice with 10 μg of immunogen adjuvanted with Sigma Adjuvant System (SAS) using a prime-boost regimen. Serum was collected one week prior to boost (week 2) and infection (week 7), respectively (Supplementary Fig. 4). We then compared the ability of post-boost sera from these

immunized mice to neutralize recombinant vesicular stomatitis virus displaying spike proteins (rVSV-CoVs) derived from a wide range of betacoronaviruses. All rVSV-CoVs displaying sarbecovirus-derived spike proteins (Wuhan-1, Omicron BA.1, SARS-CoV, and SHC014) were neutralized by sera from mice immunized with the full HexaPro spike protein ectodomain, which served as our positive control (Fig. 4a-e). Additionally, sera from mice immunized with S2 immunogens HexaPro-S2 (base construct), HexaPro-SS, HexaPro-SS-PentaStalks, and HexaPro-SS-Δstalk significantly neutralized rVSVs bearing spikes from SARS-CoV-2 Wuhan-1, SARS-CoV, and SHC014 compared to those mice immunized with PBS (Fig. 4a, c, d). Of mice immunized with S2 constructs, only HexaPro-SS-Pentastalk and HexaPro-SS-Δstalk significantly neutralized rVSV-SARS-CoV-2 Omicron BA.1 compared to mice immunized with PBS (Fig. 4b). No significant neutralization was observed for rVSV-MERS-CoV when tested with sera from mice immunized with HexaPro or any of the S2-only constructs (Fig. 4e).

## HexaPro-SS vaccinated mice are fully protected against lethal SARS-CoV-2 challenge and partially protected against lethal SARS-CoV challenge

To assess the ability of the various S2-only constructs to protect against lethal infection, we immunized 10-week-old female BALB/c mice with 10 μg of one immunogen and boosted with the same immunogen three weeks after prime immunization. Both immunizations used SAS as the adjuvant. Five weeks after the boost, animals were infected with either $10^4$ PFU SARS-CoV-2 MA10 or $10^4$ PFU of the heterologous SARS-CoV MA15[34–36] and monitored daily for changes in

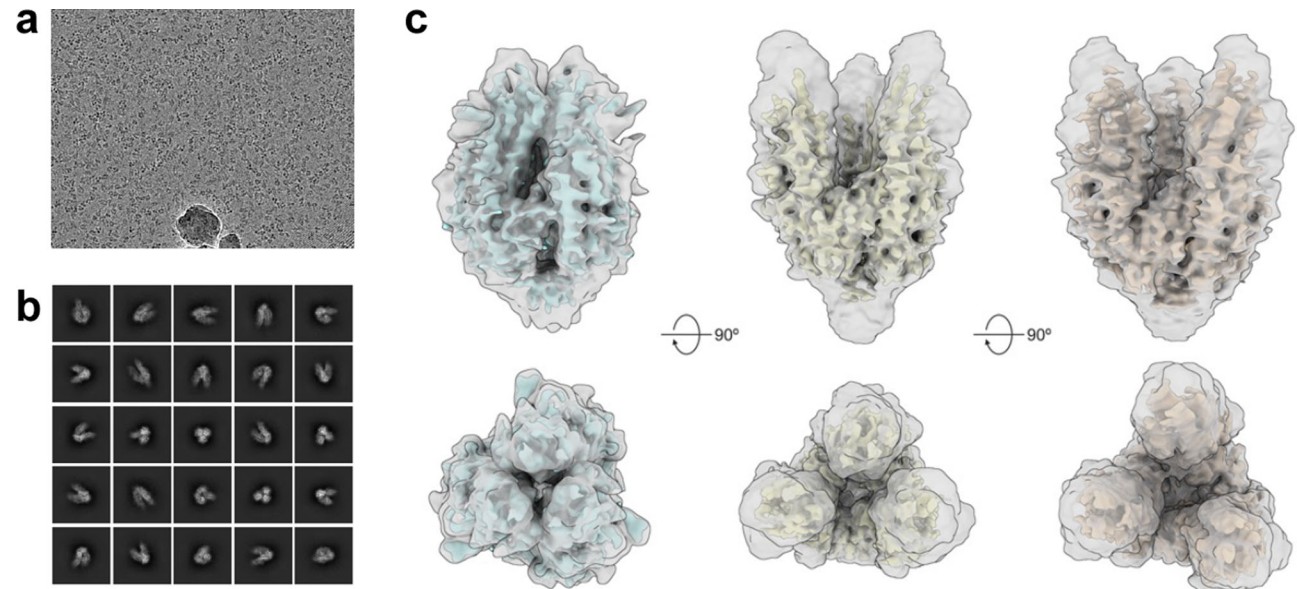

**Fig. 3 | Cryo-EM structure of disulfide-stabilized S2 reveals a range of flexibility at the trimer apex. a** A representative micrograph (from 3350 collected) and **b** 2D class averages of HexaPro-SS-Δstalk. **c** 3D heterogenous reconstructions of HexaPro-SS-Δstalk demonstrate three distinct conformations with the trimer apex closed (left), semi-open (middle), and fully open (right).

body weight and signs of morbidity. As soon as animals reach 80% of their starting weight, they are subject to more stringent observation with weights measured twice a day as well as a visual check-in between measurements. Animals that approach 70% of their starting weight are humanely euthanized. Dedicated groups of mice for each immunization regimen were sacrificed for tissue collection on days 2, 4, and 7 (Supplementary Fig. 4b). Only animals immunized with HexaPro spike ectodomain were fully protected against challenge with SARS-CoV-2 MA10, with no changes in body weight after infection (Fig. 5a). Importantly, even though some S2-only constructs provided minimal but statistically significant protection from weight loss (HexaPro-SS as well as HexaPro-SS-PentaStalks; Fig. 5a), they conferred significant protection from mortality with only one mouse succumbing to infection in the HexaPro-SS-Δstalk-immunized group. All mock-immunized but virus-challenged control mice succumbed to infection by day 4 (Fig. 5b). Gross evaluation of macroscopic changes in coloration of lung tissue at the time of sample collection (congestion score: 0–4; 0 = healthy pink lung, 1 = 25%, 2 = 50%, 3 = 75%, 4 = 100% of whole lung tissue exhibits dark red discoloration) confirmed this trend. HexaPro-immunized mice exhibited no changes in lung coloration, whereas all other vaccinated mice showed minimal changes (congestion scores: 0–1/4). In contrast, severe alterations in lung coloration were observed in the PBS-immunized control group (congestion scores: 2.5–4/4) on day 4 and no mice survived until day 7. Statistical differences were found between HexaPro-immunized mice and HexaPro-SS-Δstalk-immunized mice late during infection (day 7) (Fig. 5c). Inhibition of viral replication was obvious and statistically significant compared to PBS-immunized mice in the HexaPro-vaccinated group on day 2 after infection with a reduction of two orders of magnitude ($10^3$ PFU) in viral titer in the upper respiratory tract compared to similar titers in all other groups ($10^5$ PFU). Four days after infection, virus was cleared from nasal turbinates in the HexaPro group, whereas all other groups showed average titers around 100 to 1000 PFU. On day 7 after infection, virus was cleared in all experimental groups (Fig. 5d). The difference observed in the upper respiratory tract early during infection was even more pronounced than in the lower respiratory tract, with no detectable titer in the HexaPro group compared to lung titers between $10^6$ to $10^7$ PFU in all other immunized groups on day 2 after infection. PBS-vaccinated mice exhibited the highest lung titers at $10^7$ PFU on day

2 after infection. Viral lung titers on day 4 after infection were reduced in all immunized groups by roughly 3 logs, whereas the control group lung titer decreased by only 1 log. As observed in the upper respiratory tract, all animals surviving until day 7 after infection showed no detectable lung titers (Fig. 5e).

To test the breadth of protection provided by immunization with HexaPro and S2-only immunogens, we immunized 10-week-old female BALB/c mice as described above and challenged them with a lethal dose of SARS-CoV MA15. In contrast to the SARS-CoV-2 MA10 challenge, none of the constructs provided significant protection from weight loss after challenge with $10^4$ PFU of mouse-adapted SARS-CoV MA15 (Fig. 6a). However, all constructs conferred significant protection from mortality compared to the control group, in which all mice succumbed to infection by day 6. Among the groups of mice immunized with the different spike constructs, HexaPro, HexaPro-SS, and HexaPro-S2 showed similar protection with mortality rates of 20% (Fig. 6b). Gross evaluation of changes in lung coloration showed no significant differences by day 2 after infection (Fig. 6c). After day 2, there were no control animals left for comparison, however, the trends observed in mortality rates were confirmed via immunization with HexaPro, HexaPro-SS, and HexaPro-S2 leading to less discoloration of lung tissue compared to immunization with HexaPro-SS-Δstalk and HexaPro-SS-PentaStalks. Significant differences were found on day 4 between HexaPro-SS-Δstalk and mock-infected control groups and on day 7 between HexaPro-S2-immunized as well as HexaPro-SS-Δstalk-immunized mice and mock-infected control mice (Fig. 6c).

Viral titers in the upper respiratory tract (Fig. 6d) were similar across all groups at around $10^5$ PFU on day 2 after infection and $10^4$ PFU on day 4 after infection. All groups showed similarly high viral titers in the lower respiratory tract at $10^7$ PFU by day 2 after infection, which decreased to $10^4$ by day 4. The only exceptions were the two surviving mice in the HexaPro-SS-PentaStalks immunized group exhibiting slightly elevated titers ($10^5$ PFU) in the upper respiratory tract on day 4 as well as the HexaPro-SS-immunized group showing reduced lung titers ($10^4$ PFU) on day 4 after infection (Fig. 6e). By day 7 after infection neither upper nor lower respiratory tract samples showed any detectable titers. These results demonstrate that S2-only immunogens can provide protection against SARS-CoV-2 and partial protection

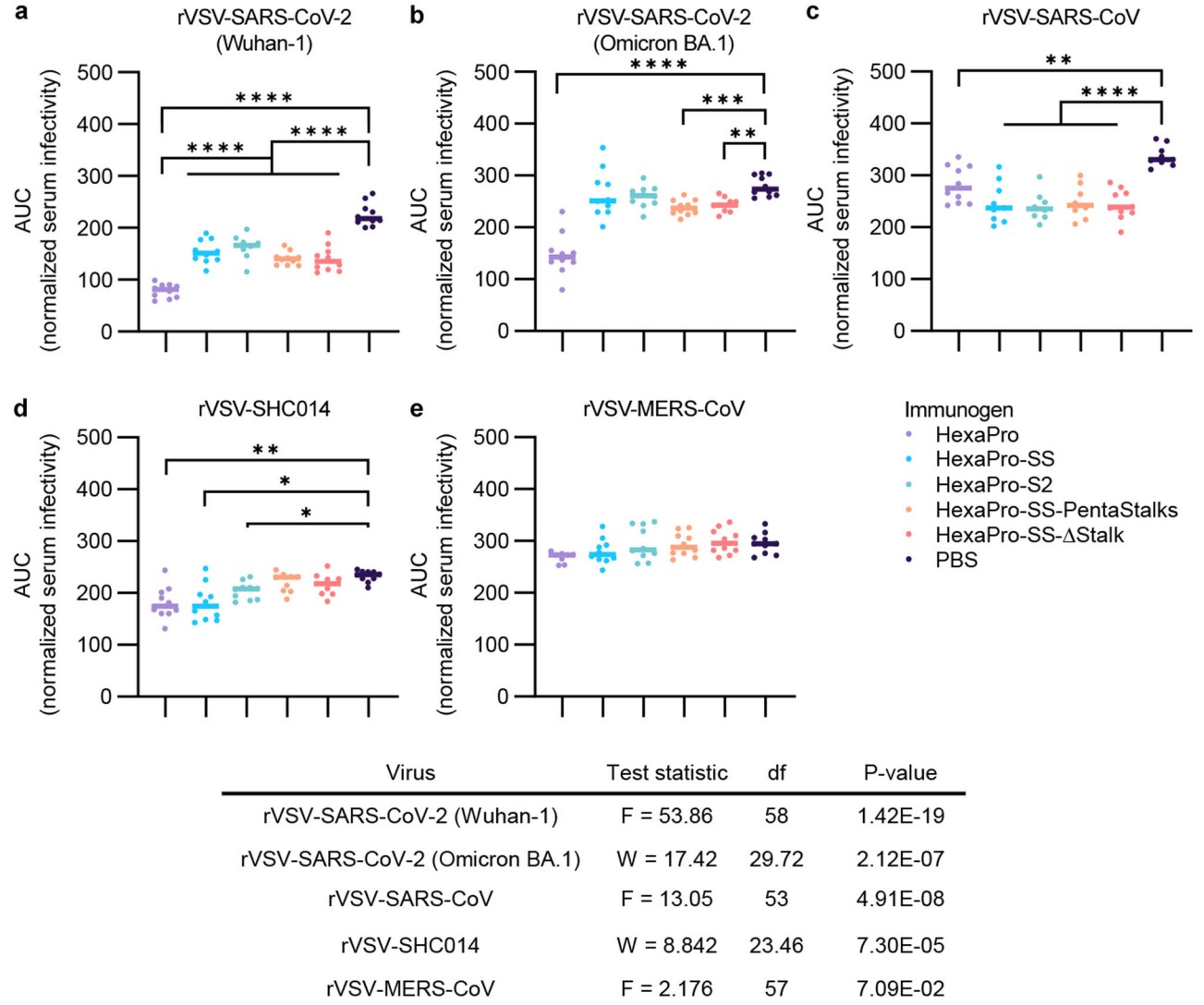

| Virus | Test statistic | df | P-value |
|---|---|---|---|
| rVSV-SARS-CoV-2 (Wuhan-1) | F = 53.86 | 58 | 1.42E-19 |
| rVSV-SARS-CoV-2 (Omicron BA.1) | W = 17.42 | 29.72 | 2.12E-07 |
| rVSV-SARS-CoV | F = 13.05 | 53 | 4.91E-08 |
| rVSV-SHC014 | W = 8.842 | 23.46 | 7.30E-05 |
| rVSV-MERS-CoV | F = 2.176 | 57 | 7.09E-02 |

**Fig. 4 | Sera from mice immunized with S2 immunogens can neutralize a broad range of rVSV-CoVs. a-e** Pre-titrated amounts of rVSV-CoVs were incubated with serial 3-fold dilutions of sera from mice ($n = 10$/immunized group) immunized with respective S antigens or PBS at RT for 1 hr. Virus-sera mixtures were then added to monolayers of Vero cells. At 10 hr post-infection, cells were fixed, and nuclei were counterstained. Infected cells were scored by BioTek Cytation5 for presence of GFP. Sera from 10 mice are included in each group. Area under the curve (AUC) was calculated from normalized infectivity levels. One-way ANOVA with Tukey's multiple comparisons test (**a, c, e**) or Welch's ANOVA with Dunnett's T3 multiple comparisons test (**b, d**) were run based on normality and homoscedasticity. **** $p < 0.0001$, *** $p < 0.001$, ** $p < 0.01$, * $p < 0.05$ **a** PBS:HexaPro, HexaPro-SS, HexaPro-S2, HexaPro-SS-PentaStalks, Hexapro-SS-Δstalk $p = <1e-15$, 2.86e-9, 2.61e-7, 1.48e-11, 1.11e-11, HexaPro:HexaPro-SS, HexaPro-S2, HexaPro-SS-PentaStalks, Hexapro-SS-Δstalk $p = 5.64e-10$, 3.01e-11, 1.04e-7, 1.34e-7; **b** PBS:HexaPro-SS-PentaStalks, Hexapro-SS-Δstalk $p = 7.37e-4$, 2.40e-3; **c** PBS:HexaPro, HexaPro-SS, HexaPro-S2, HexaPro-SS-PentaStalks, Hexapro-SS-Δstalk $p = 5.14e-3$, 2.75e-6, 9.77e-7, 6.54e-6, 6.68e-7; **d** PBS:HexaPro, HexaPro-SS, HexaPro-S2 $p = 6.20e-3$, 1.31e-2, 7.20e-3. Source data are provided as a Source Data file.

against SARS-CoV despite not being able to elicit highly neutralizing RBD- and NTD-directed antibodies.

## Double HexaPro boost increases protection against SARS-CoV in BALB/c mice

Based on the results of BALB/c mice immunized with different spike constructs showing protection from mortality but not weight loss in a lethal SARS-CoV MA15 challenge (Fig. 6), we altered our immunization strategy in an attempt to increase beneficial effects on cross-protection. Therefore, we immunized 10-week-old female BALB/c mice with 10 μg HexaPro, deglycosylated HexaPro-SS, or PBS (week 0) followed by a first boost (week 6) with either HexaPro, HexaPro-SS, deglycosylated HexaPro-SS ('degly'), or PBS followed by a second boost (week 10) with the same antigens as in the first boost (Fig. 7a). Given that glycosite-deleted spike protein elicited a broad-spectrum humoral and cell-mediated immunity[37], we hypothesized that

deglycosylated HexaPro-SS could also provide a similar advantage. Thus, we treated HexaPro-SS with Endo H, leaving a single GlcNAc at each N-linked glycosylation site on S2. To see whether we could enhance the breadth of neutralization against a range of rVSV-CoVs, microneutralization assays were performed with mouse sera collected 2 weeks after the third dose of immunogen. We found that mouse sera from all four immunization strategies significantly neutralized sarbecovirus rVSV-CoVs rVSV-SARS-CoV-2 Wuhan-1, rVSV-SARS-CoV, and rVSV-SHC014 compared to mice immunized with PBS (Supplementary Fig. 5a, c, d). Both triple HexaPro immunization and HexaPro followed by 2 immunizations of deglycosylated HexaPro-SS significantly neutralized rVSV-SARS-CoV-2 Omicron BA.1 compared to PBS immunized mouse sera, with triple HexaPro immunization being more strongly neutralizing ($p < 0.0001$) (Supplementary Fig. 5b). There was a modest, but significant neutralization of rVSV-MERS-CoV seen with sera from mice immunized with HexaPro followed by 2 shots of deglycosylated

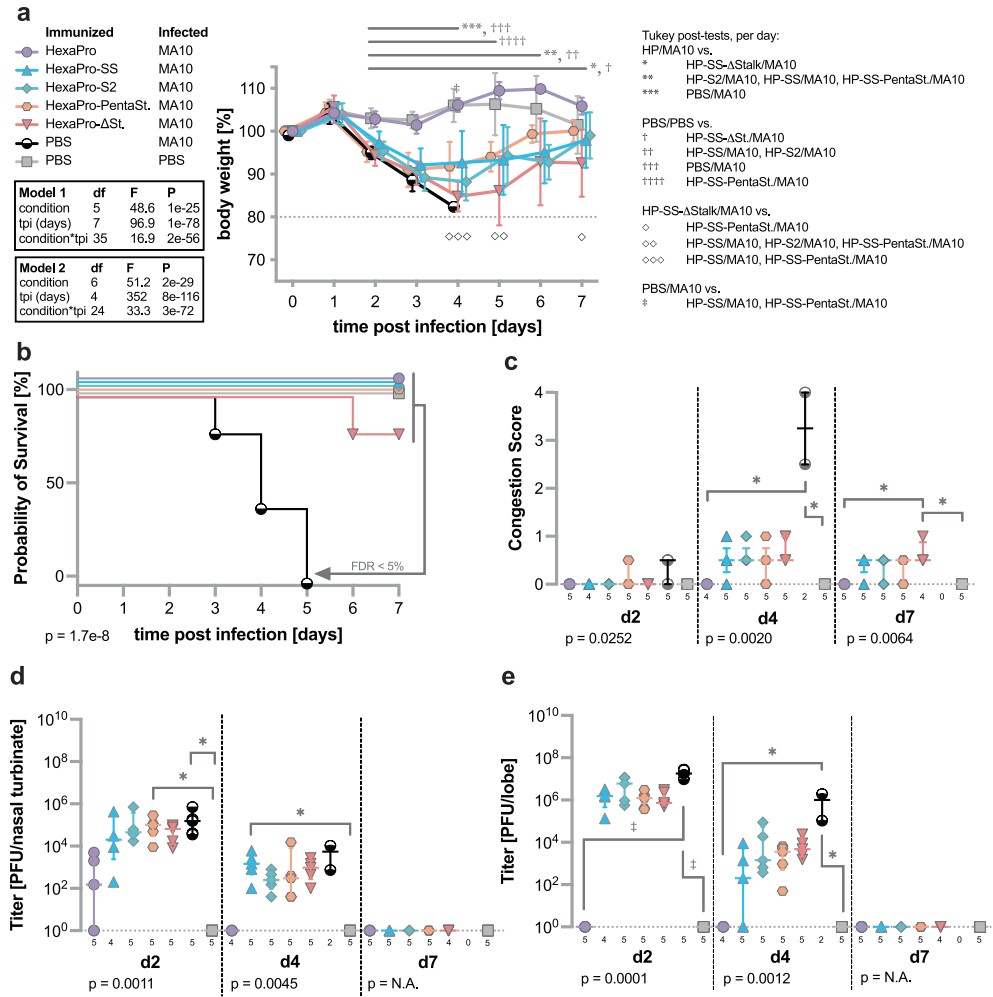

**Fig. 5 | HexaPro-SS provides full protection against SARS-CoV-2 challenge.**
After being immunized with respective S antigens or PBS, BALB/cAnNHsd mice were challenged with a lethal dose of mouse-adapted SARS-CoV-2 ($n = 14$ mice for HexaPro and HexaPro-SS, $n = 15$ mice for all other groups). **a**, Mouse body weights were monitored daily following challenge, and the percentage of body weight loss over time is represented by a line plot summarized per strain per day by the mean ± SD. We analyzed the change in body weight using mixed models with repeated measures (exact $P$ values for each term and the interaction is reported in the plot) followed by Tukey post-tests ($n = 14$ mice for HexaPro and HexaPro-SS, $n = 15$ mice for all other groups at day 0). Per-day comparisons with post-tests <0.05 are indicated with symbols defined in the key. **b**, The survival graph

represents the probability of survival over two to seven days post-challenge ($n = 5$ mice per group). The Mantel-Cox log-rank test determined the survival curves were different ($\chi^2 = 47.18$, df = 6, $P = 1.7e$-8) and the two-stage linear step-up procedure of Benjamini, Krieger and Yekutieli identified the pairwise differences of survival curves with a false discovery rate (Q) < 5%. **c**, The congestion score in lung and the viral titers in **d**, nasal turbinate and **e**, lung ($n$ per group listed in the graphs) are represented by dot plot, summarized by the median ± IQR. We analyzed the congestion scores and titers using ANOVA on ranks per day with each $P$ value reported in the figure. Dunn's pairwise comparison post-tests for $p < 0.05$ and $p < 0.001$ are noted by * and ‡, respectively. Source data are provided as a Source Data file.

HexaPro-SS ($p < 0.05$) compared to mice immunized with PBS (Supplementary Fig. 5e).

To investigate whether this immunization strategy can protect the mice from SARS-CoV, double-boosted mice were infected 8 weeks after the second boost (week 18) with $10^4$ PFU mouse-adapted SARS-CoV MA15 and monitored daily for clinical signs of disease. Dedicated groups of mice for each immunization regimen were sacrificed for tissue collection on days 2 and 4 after infection (Fig. 7a). To reduce morbidity and mortality in our experimental cohort, we decided to end the experiment on day 4 instead of day 7 to keep the number of animals that reach experimental endpoints for humane euthanasia as low as possible. Only triple HexaPro-immunized mice showed significant protection from weight loss whereas all other groups followed the control group trajectory (Fig. 7b). However, all but the HexaPro-primed, HexaPro-SS-boosted immunization strategy conferred protection from mortality until the end of the study at day 4 (Fig. 7c). No significant differences were observed during gross evaluation of lung

discoloration at the time of sample collection on day 2 after infection. Maximal congestion score was recorded for the one remaining control animal on day 4 after infection, whereas all immunized mice exhibited congestion scores of 0.5 to 2 with double HexaPro-boosted animals showing the least amount of change in lung coloration (Fig. 7d). Average viral titers in upper respiratory tracts were similar across all groups on both harvest days. Importantly, only two HexaPro double-boosted animals showed viral clearance on day 4 after infection (Fig. 7e).

Significant differences in viral titers in the lower respiratory tract were observed on day 2 after infection with lower titers in the HexaPro-HexaPro-HexaPro and degly-degly-degly immunized groups in comparison to mock-immunized animals. However, on day 4 we detected a clear difference in lung titers between the one surviving control animal ($10^6$ PFU) and all immunized groups ($10^2$ to $10^4$ PFU). Again, 3 out of 5 animals from the double HexaPro-boosted group were able to clear virus by day 4. Additionally, 1 out of 5 animals from the double

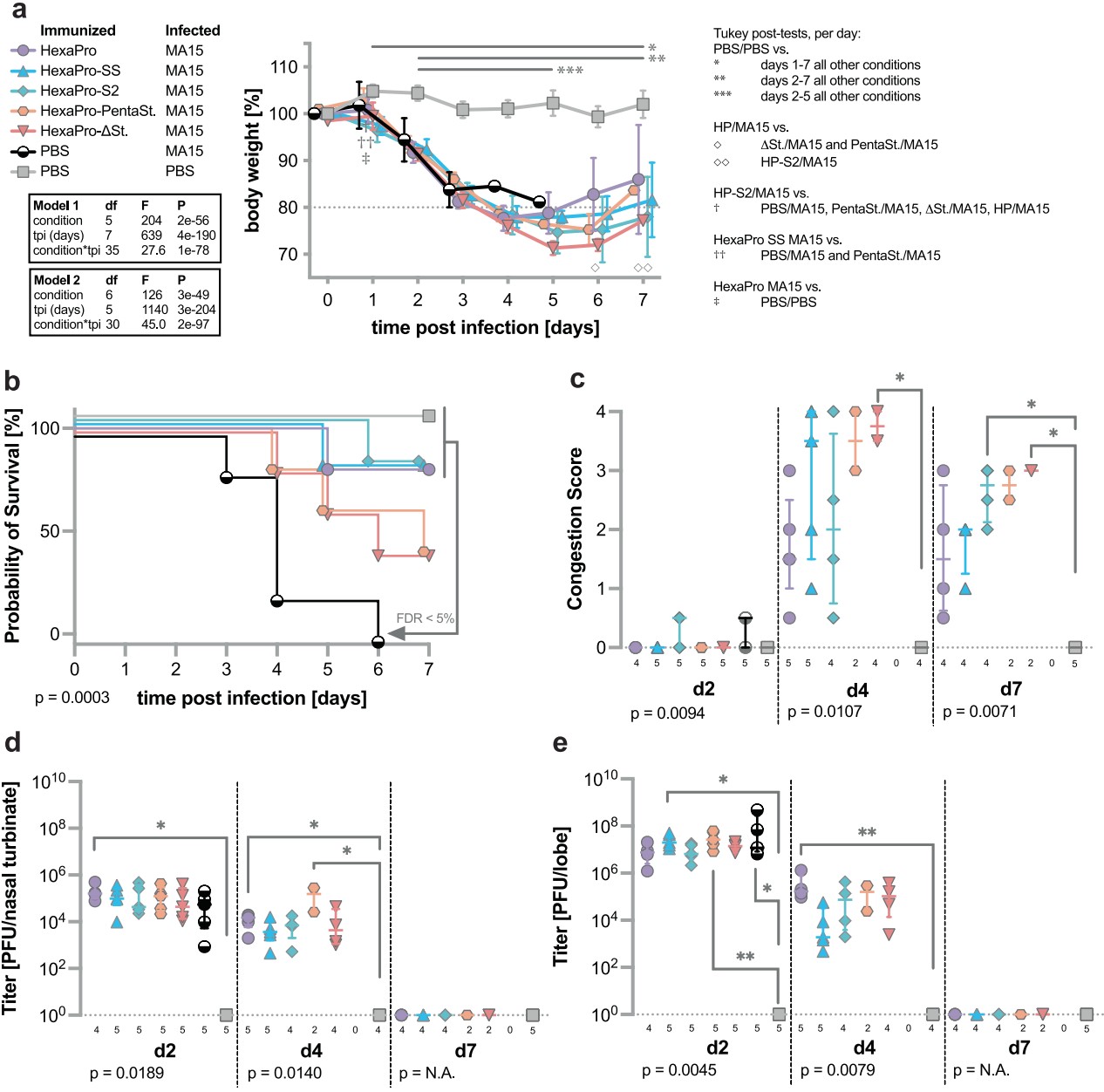

**Fig. 6 | HexaPro-SS provides partial protection against SARS-CoV challenge.** After being immunized with respective S antigens or PBS, BALB/cAnNHsd mice were challenged with a lethal dose of mouse-adapted SARS-CoV ($n = 14$ mice for HexaPro and PBS PBS, $n = 15$ mice for all other groups). **a** Mouse body weights were monitored daily following challenge, and the percentage of body weight loss over time is represented by a line plot summarized per strain per day by the mean ± SD. We analyzed the change in body weight using mixed models with repeated measures (exact $P$ values for each term and the interaction is reported in the plot) followed by Tukey post-tests ($n = 14$ mice for HexaPro and PBS PBS, $n = 15$ mice for all other groups at day 0). One animal from the HexaPro-immunized group as well as one animal from the HexaPro-SS-immunized group reached 70% of their starting weight on day 7 after infection. Per-day comparisons with post-tests <0.05 are indicated with symbols defined in the key. **b**, The survival graph represents the probability of survival over two to seven days post-challenge ($n = 5$ mice per group). The Mantel-Cox log-rank test determined the survival curves were different ($\chi^2 = 25.36$, df = 6, $P = 3e\text{-}4$) and the two-stage linear step-up procedure of Benjamini, Krieger and Yekutieli identified the pairwise differences of survival curves with a false discovery rate (Q) < 5%. **c** The congestion score in lung and the viral titers in **d**, nasal turbinate and **e**, lung ($N$ per group listed in the graphs) are represented by dot plot, summarized by the median ± IQR. We analyzed the congestion scores and titers using ANOVA on ranks per day with each $P$ value reported in the figure. Dunn's pairwise comparison post-tests for $p < 0.05$ and $p < 0.01$ are noted by * and ***, respectively. Source data are provided as a Source Data file.

deglycosylated boosted group exhibited no detectable lung titer (Fig. 7f). Collectively, these data show that only a double boost with HexaPro provides increased protection not only from mortality but also from changes in body weight after SARS-CoV infection compared to a single boost (Fig. 6 and Fig. 7). Vaccination approaches with HexaPro in combination with two boosts of each construct did not protect from clinical signs of disease.

## Discussion

The reduced effectiveness of current licensed vaccines and FDA-approved therapeutic antibodies against the newly emerging SARS-CoV-2 omicron subvariants has created an urgent need to develop coronavirus vaccines that provide broader breadth of immunity. Multivalent display of RBDs from different sarbecoviruses on the nanoparticle platform[38–40] and multivalent cocktails of spike chimera

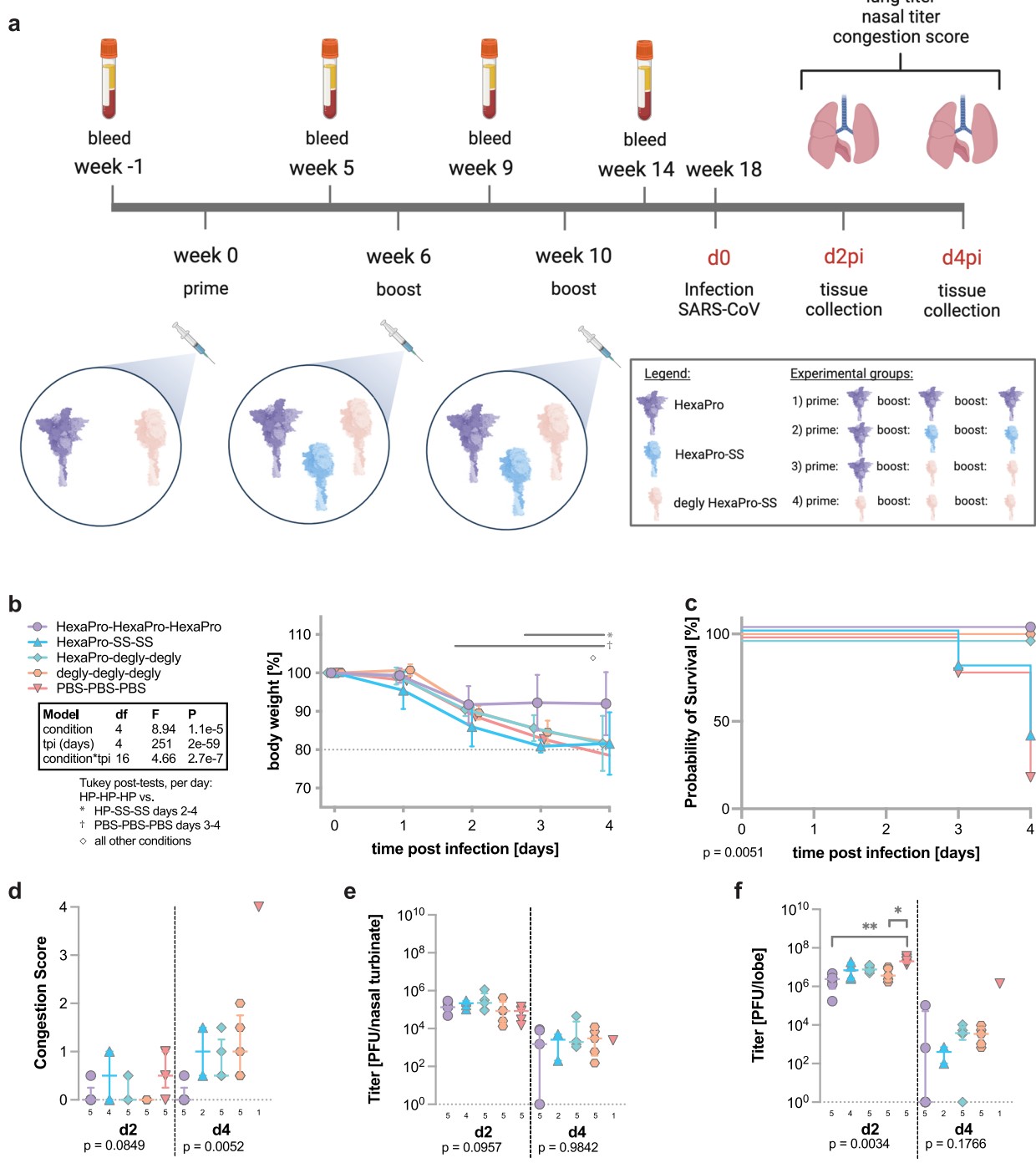

**Fig. 7 | Double HexaPro boost provides best cross-protection against SARS-CoV challenge. a** After being immunized with respective S antigens or PBS, BALB/cAnNHsd mice were challenged with a lethal dose of mouse-adapted SARS-CoV MA15 (*n* = 9 mice for HexaPro-SS-SS, *n* = 10 mice for all other groups). The cartoons were created by BioRender.com. **b** Mouse body weights were monitored daily following challenge, and the percentage of body weight loss over time is represented by a line plot summarized per strain per day by the mean ± SD. We analyzed the change in body weight using mixed models with repeated measures (exact P values for each term and the interaction are reported in the plot) followed by Tukey post-tests (*n* = 9 mice for HexaPro-SS-SS, *n* = 10 mice for all other groups at day 0). Per-day comparisons with post-tests <0.05 are indicated with symbols

defined in the key. **c** The survival graph represents the probability of survival over two to four days post-challenge (*n* = 5 mice per group). The Mantel-Cox log-rank test determined the survival curves were different ($\chi^2$ = 14.48, df = 4, *P* = 5.9e-3). The two-stage linear step-up procedure of Benjamini, Krieger and Yekutieli did not identify any pairwise differences of survival curves with a false discovery rate (Q) < 5%. **d** The congestion score in lung and the viral titers in **e**, nasal turbinate and **f**, lung (*n* per group listed in the graphs) are represented by dot plot, summarized by the median ± IQR. We analyzed the congestion scores and titers using ANOVA on ranks per day with each *P* value reported in the figure. Dunn's pairwise comparison post-tests for *p* < 0.05 and *p* < 0.01 are noted by * and **, respectively. Source data are provided as a Source Data file.

delivered via mRNA[41] have demonstrated promising results against diverse betacoronaviruses in animal models. Instead of using a variety of antigens, we focused our efforts on solving complex structural problems associated with engineering the sequence-conserved SARS-CoV-2 S2 subunit[42] (Supplementary Fig. 6). The full-length S ectodomain transiently samples an open conformation that exposes cryptic S2 epitopes[43]. We hypothesized that without the S1 subunit and its intertwined RBDs functioning as a natural trimerization motif, S2-only trimers would readily separate into individual protomers. Therefore, with the entire S1 subunit removed, we introduced interprotomer disulfide bonds to lock the S2 subunit in a prefusion trimer, with the intention of eliciting broadly neutralizing antibodies. We also incorporated tandem stem helix regions from four other betacoronavirus (HexaPro-SS-PentaStalks), with the goal of inducing higher titers of stem helix-specific antibodies[16–21]. Moreover, we structurally characterized HexaPro-SS-Δstalk, a stabilized S2 lacking the flexible C-terminal stalk, which revealed that it adopts the prefusion conformation. Immunization with a variety of stabilized S2 antigens elicited antibodies with cross-neutralization activity to several sarbecoviruses. Notably, HexaPro-SS, even lacking the immunodominant S1 subunit, protected mice from SARS-CoV-2 lethal challenge (Fig. 5) and partially protected mice from SARS-CoV challenge (Fig. 6), comparable to full-length S ectodomain.

There is growing evidence that the membrane-distal apex of class I fusion proteins, such as influenza A virus HA[44] and the fusion proteins from RSV and hMPV[45,46], open or breathe in a way that allows internal surfaces to become transiently accessible to the immune system[47–50]. Here, via structural studies, we demonstrate that the apex of our disulfide-stabilized S2 antigens exists in a range of prefusion conformations: from closed to splayed open. These data are consistent with published HDX-MS experiments that also indicated that the S2 apex (residues 962–1024) becomes solvent accessible even in the context of full-length spike ectodomain[43]. This transient breathing of class I viral fusion proteins likely occurs in vivo on the surface of virions, as antibodies against internal epitopes have been isolated from individuals as a result of natural infection[47]. A class of dominant public antibodies enriched following SARS-CoV-2 infection recognize a cryptic epitope on the S2 apex[51]. Similarly, antibody 3A3, broadly reacting to MERS-CoV, SARS-CoV, and SARS-CoV-2 S, binds to a highly conserved region on the S2 apex[52]. Notably, these S2 antibodies preferentially bind to a conformation of the spike when the S2 apex is open[43,51,52]. For other respiratory viruses, two types of human antibodies recognizing the interior surfaces of hMPV prefusion F were also recently discovered[47,48]. FluA-20 is another example of a human antibody that binds to cryptic loops within the influenza A HA head and further opens the trimer[49]. Many of these weak-to-non-neutralizing internal binders represent a dominant population in the human B cell repertoire[47,51]. Therefore, advanced designs to fully stabilize the prefusion trimer interface could be a viable option to potentially improve the immunogenicity of SARS-CoV-2 S2-only vaccine antigens.

Few studies have investigated the immunogenicity of S2 or parts of S2 from SARS-CoV-2 in preclinical animal models. Ng et al., showed that S2-targeted vaccination (via DNA) could elicit broadly neutralizing antibody responses and protect the mice against SARS-CoV-2 infections with the parental and alpha variant[53]. However, the S2 recombinant protein used in their study failed to induce neutralizing antibodies. Similar results were also found in a prior study using SARS-CoV-2 S2 subunit as an antigen[54]. It is highly likely that those recombinant S2 antigens were not well-folded because stabilizing mutations were not introduced to maintain S2 in a prefusion conformation and eukaryotic expression systems were not used to produce the antigen. Moreover, non-neutralizing antibodies that target S2 protect against sarbecovirus challenge[21,22]. Here, we were able to use prefusion-stabilized S2 to elicit broadly neutralizing antibodies against SARS-CoV-2 VOCs and other sarbecoviruses with crossover potential. We

also primed with full S ectodomain and then boosted with recombinant S2 protein to mimic the current situation in the human population where the majority of people have been vaccinated. Surprisingly, S primed- and deglycosylated S2-boosted mice generated antibody responses that could weakly neutralize MERS-CoV. It is possible that this cross-neutralization might result from stem-helix- or fusion-peptide-targeting antibodies. Further investigation using isolated monoclonal antibodies are needed to validate this hypothesis.

Among all S2 constructs, HexaPro-SS-Δstalk elicited inferior protection than others against SARS-CoV-2 and SARS-CoV challenges, indicating that stem-helix epitopes are important for favorable vaccine responses. This is consistent with positive findings from peptide-based[55] and ferritin nanoparticle[56] vaccines using the S2 region containing the stem helix. Although our S2 constructs offered protection against SARS-CoV-2 challenge, the full protection did not extend to 10-LD50-fold higher lethal doses of SARS-CoV. It is likely the S2-only antigen did not elicit robust antibody responses against protective epitopes, particularly the fusion peptide and the stem helix. In addition, stem-helix antibodies targeting these regions are rare in human B cell repertoires and require somatic maturation to increase affinity and neutralization potency[18]. In contrast with HA stem-directed antibodies[57,58], the epitopes of these S2-targeted antibodies are only transiently exposed when S2 is transitioning to the intermediates and postfusion conformation[23,51,52]. Future S2 antigen design should aim to elicit antibodies with high affinity to the external surface of spike in the prefusion conformation.

Collectively, we demonstrated that our prefusion-stabilized S2-only antigens could elicit broadly neutralizing responses to several sarbecoviruses and protect mice from SARS-CoV-2 challenge. A limitation of this study is that we did not assess the effector functions of the elicited antibodies[59], which in combination with antibody neutralization likely contribute to the observed protection. Antibodies that target membrane proximal regions of the related fusion proteins from influenza and HIV-1 appear to induce stronger antibody-dependent cellular cytotoxicity and complement deposition[60–64], which could help ameliorate disease. Cross-reactive S2-specific antibodies were also found to be more abundant in survivors of COVID-19 compared to non-survivors[65]. It has been recently shown that non-neutralizing antibodies targeting HR1, fusion peptide, and HR2 can stimulate FcγR4 effector functions and convey protection to diverse sarbecoviruses[22]. Thus, it will be interesting to structurally characterize the polyclonal antibody responses from the S2-immunized mice via EM[66,67] and investigate the epitopes targeted by the elicited antibodies. Functional epitopes that are recognized by neutralizing antibodies and the antibodies associated with potent effector functions will inform additional antigen design efforts. For future investigation, updating the constructs to currently circulating SARS-CoV-2 variants, such as BA.5, XBB1.5 and BA.2.86, may confer increased protection. Furthermore, masking non-neutralizing epitopes with N-linked glycans, controlling the apex breathing with minimal exposure of cryptic epitopes, and displaying multiple strains of S2 on a nanoparticle platform could all be viable options to develop next-generation pan-coronavirus vaccines.

## Methods
### Protein expression and purification
The base construct HexaPro-S2 used for the SARS-CoV-2 S2 variants contained residues 697–1208 of SARS-CoV-2 S (GenBank ID: MN908947) with proline substituted at residues 817, 892, 899, 942, 986, and 987, and C-terminal foldon trimerization motif of T4 fibritin, an HRV3C protease recognition site, an 8xHis tag, and a Strep-tag II, cloned into the mammalian expression plasmid pαH. The disulfide variants were designed based on Cβ distances of the paired residues, and may contain one, two, or three pairs of Cys substitutions. The additional proline (Q895P) and charged substitution (Q957E) were

chosen because both boosted SARS-CoV-2 S expression[30]. The Δstalk construct had residues 1142–1208 deleted from spike ectodomain. In addition to its own helical stalk (SARS-CoV-2 S residues 1141–1161), the PentaStalks construct contained corresponding helical stalk regions from OC43 (residues 1233–1246), HKU1 (residues 1234–1247), MERS-CoV (residues 1231–1244), and HKU-9 (residues 1147–1160), while TriStalks only contained helical stalk regions from MERS-CoV and HKU1. Plasmids encoding S or S2 variants were used to transfect FreeStyle 293-F cells (ThermoFisher, Cat #R79007) with poly-ethylenimine (PEI). Kifunensine was added to a final concentration of 5 μM and, for large-scale transfections, pluronic F-68 was added to a final concentration of 0.1% v/v. The supernatant was harvested 6 days after transfection and then applied to a StrepTactin column (IBA) for affinity purification. For obtaining trimeric spikes with higher purity, the elu-tion from the StrepTactin column was applied to a Superose 6 Increase 10/300 or Superose 6 16/70 size-exclusion column (SEC) (GE Health-care) in SEC buffer (2 mM Tris pH 8.0, 200 mM NaCl, and 0.02% NaN₃). To generate deglycosylated HexaPro-SS-Δstalk and deglycosylated HexaPro-SS, affinity-purified proteins were treated with Endo H at 4 °C overnight prior to the SEC purification. The 8xHis tag and Strep-tag II were also removed by treating the proteins with HRV3C protease at 4 °C overnight prior to the SEC purification. The SEC peak fractions were pooled and concentrated to a higher concentration for structural studies and for animal immunization.

### Differential scanning fluorimetry
Purified SARS-CoV-2 S2-only variants at a concentration of 1 μM were mixed with a final concentration 5X SYPRO Orange Protein Gel Stain (Thermo Fisher) in a white, opaque 96-well plate. Continuous fluor-escence measurements ($\lambda ex = 465$ nm, $\lambda em = 580$ nm) were conducted using a Roche LightCycler 480 II, with a temperature ramp rate of 4.4 °C/minute, and a temperature range of 25 °C to 95 °C. Data were plotted as the derivative of the melting curve.

### Biolayer interferometry
A set of 8 anti-hIgG Fc Capture (AHC) Biosensors (Sartorius, cat # 18-5064) were pre-wet in 1x HBS-EP+ buffer (Cytiva, Cat # BR100669) for 10 minutes at room temperature and then loaded on to Octect RED96e (Forte´Bio). All pre-wet biosensors first underwent 60 seconds of baseline in HBS-EP+ buffer and then were dipped into 200 uL of 25 nM IgGs diluted in HBS-EP+ to capture to a level of 0.6 nm each. IgG G4[68] was used as an isotype control. All 8 AHC biosensors then underwent another 60 seconds of baseline in buffer followed by a 300-second association in 200 uL of 100 nM SARS-CoV-2 spike antigens (SARS-CoV-2 S-2P, HexaPro, HexaPro-SS, or HexaPro-SS Δstalk) and another 300-second dissociation back into buffer wells. Data were then refer-ence subtracted (IgG G4) using the software Data Analysis 11.1. Pro-cessed data were fit globally to a 1:1 binding model for both association and dissociation.

### X-ray crystallography
Crystals were initially grown by hanging-drop vapor diffusion by mixing 500 nL of HexaPro-SS-Δstalk (6.4 mg/ml) with 500 nL of reservoir solution containing 0.1 M HEPES pH 7.5, 8% (v/v) ethylene glycol, and 10% (v/v) PEG 8000. The crystal that diffracted to 3.2 Å at SBC beamline 19ID (Advanced Photon Source, Argonne National Laboratory) was produced from the same initial condition plus 4% (v/v) polypropylene glycol P400. Data were indexed and integrated in iMOSFLM[69], before being merged and scaled using Aimless[70]. Data processing strongly indicated that the crystals were twinned, in space group *R*3:H. Molecular replacement was performed in Phaser[71] using HexaPro S (PDB ID: 6XKL) as a search model. The models were then subjected to multiple rounds of model building and refinement in Coot[72], Phenix[73], and Isolde[74]. Twin law (k, h, -l) was applied during Phenix refinement, which substantially improved the maps and

$R_{work}/R_{free}$ values. Data collection and refinement statistics can be found in Supplementary Table 2.

### Cryo-EM sample preparation, data collection and processing
Purified HexaPro-SS-Δstalk at 0.55 mg/mL in 2 mM Tris pH 8.0, 200 mM NaCl, and 0.02% NaN₃ was deposited on a plasma-cleaned UltrAuFoil 1.2/1.3 grid prior to being blotted for 4 seconds with -2 force in a Vitrobot Mark IV (ThermoFisher) and plunge-frozen into liquid ethane. A total of 4,527 micrograph movies were recorded from a single grid using a Titan Krios (ThermoFisher) equipped with a K3 direct electron detector (Gatan). Data were collected at a magnifica-tion of 105,000x, corresponding to a calibrated pixel size of 0.81 Å/pix. CryoSPARC Live v4.2.1[75] was used for patch motion correction, CTF estimation, micrograph curation, particle picking, and rounds of 2D classification. The selected particles were then imported into cryoS-PARC v4.2.1 and used for ab-initio 3D reconstruction with 5 different classes. The final reconstructions of 3 different classes were generated via heterogeneous refinement and non-uniform homogeneous refinement. A full description of the data collection parameters can be found in Supplementary Table 3.

### Cell culture
Vero cells (ATCC, CCL-81), were cultured in Dulbecco's modified Eagle's medium (DMEM, high glucose; Gibco) supplemented with 2% heat-inactivated fetal bovine serum (FBS; Bio-Techne), 1% penicillin-streptomycin (Thermo Fisher), and 1% GlutaMAX (Thermo Fisher).

### Generation of rVSV-CoVs
Recombinant vesicular stomatitis viruses (rVSV) bearing the CoV spike of interest were generated as previously described[76]. A plasmid encoding the VSV antigenome was modified to replace its native gly-coprotein G with the CoV spike of interest. CoV spike sequences were obtained from GenBank (SARS-CoV-2 Wuhan GenBank MN908947.3, SARS-CoV-2 BA.1 GenBank UFO69279.1, SARS-CoV GenBank NC004718.3, MERS-CoV GenBank YP009047204.1, SHC014 GenBank AGZ48806.1). A 19 amino acid C-terminal deletion previously descri-bed in rVSV-SARS-CoV-2[76] was made in each CoV spike. An eGFP reporter gene was included in the first genome position as a separate transcriptional unit. Plasmid-based rescue of the rVSVs was carried out as previously described[76,77]. Briefly, 293FT cells were transfected using polyethylenimine with the VSV antigenome plasmid along with helper plasmids expressing T7 polymerase and VSV N, P, M, G, and L proteins. Supernatants from transfected cells were transferred to Vero cells at 48 hours post-transfection. The appearance of eGFP-positive cells confirmed the presence of virus. Virus was plaque-purified on Vero cells and then propagated by cell subculture. RNA was isolated from viral supernatants of plaque-purified virus and Sanger sequencing was performed to confirm the S gene sequence. Viral stocks were con-centrated via ultracentrifugation in an SW28 rotor at 141,000 x g for 4 hours. Virus was aliquoted and stored at -80 °C. The generation of all rVSV-CoVs and their use in tissue culture was performed at biosafety level 2 and was approved by the Environmental Health and Safety Department and Institutional Biosafety Committee at Albert Einstein College of Medicine.

### rVSV-CoV neutralization assay
Sera from mice immunized with S2 antigens were obtained 2 weeks post-boost. A 1:10 dilution of mouse sera was made followed by serial 3-fold dilutions in DMEM with 2% FBS. Mouse sera were incubated with rVSV-CoV (MERS-CoV S, SARS-CoV S, SARS-CoV-2 (Wuhan-1) S, SARS-CoV-2 (Omicron BA.1) S or SHC014 S) for 1 hour at room temperature. Media were removed from Vero cells in 96-well plate (Corning) and 40 μl virus/sera mixture was added to the well. The cells were incu-bated at 37 °C and 5% CO₂ for 10 hours for rVSV-MERS-CoV, SARS-CoV, and SARS-CoV-2 or 14–16 hours for rVSV-SHC014. The cells were

then fixed with 4% paraformaldehyde, washed with 1X PBS and stored in 1X PBS with Hoechst 33342 (Invitrogen) at a dilution of 1:2000. Viral infection was quantified by automatic enumeration of GFP-positive cells from captured images using Cytation 5 automated fluorescence microscope (BioTek) with analysis in Gen5 data analysis software (BioTek). The area under the curve (AUC) of the sera was calculated using GraphPad Prism software. Each sera group was tested for normality and homoscedasticity using Anderson-Darling test and Bartlett's test, respectively. Parametric and non-parametric tests were run accordingly. All statistical testing was performed using GraphPad Prism software.

### Immunization and challenge experiments

**Single boost.** Female BALB/cAnNHsd (BALB/c) mice (10 weeks old at prime immunization; Envigo: #047) were immunized with 10 μg of various spike antigen constructs adjuvanted with Sigma Adjuvant System (SAS) and boosted four weeks after in the same fashion. Constructs were diluted in PBS to a total volume of 25 μl and 25 μl of adjuvant were added to obtain a total of 50 μl per mouse. 25 μl were given intramuscularly in each hind leg. Mice were bled one week prior to prime, boost, and challenge. Serum was collected from each mouse and used in rVSV-CoV microneutralization assays. In week 8 after prime, mice were moved into the BSL3 laboratory and challenged after acclimation.

All mice (n = 5/treatment group/harvest day) were intranasally infected with $10^4$ plaque-forming units (PFU) of mouse-adapted SARS-CoV-2 MA10 or $10^4$ PFU of mouse-adapted SARS-CoV MA15 while anesthetized with a mixture of Ketamine/Xylazine. Animals were monitored daily for changes in body weight and clinical signs of disease. At indicated harvest time points (days 2, 4, and 7 after infection) mice were sacrificed, macroscopic changes in lung coloration (congestion scores) recorded, and lung tissue harvested to determine viral load. Additionally, nasal turbinates were collected to evaluate viral load in the upper respiratory tract.

**Double boost.** Female BALB/cAnNHsd (BALB/c) mice (10 weeks old at prime immunization; Envigo: #047) were immunized with 10 μg of HexaPro or deglycosylated HexaPro adjuvanted with Sigma Adjuvant System (SAS) and boosted six weeks after with various S2 constructs. The second boost was administered 4 weeks after the first boost with the same antigen. Serum from each animal was collected one week prior to prime and each boost as well as 1 week prior to challenge. Mice were infected with $10^4$ PFU of mouse-adapted SARS-CoV MA15 8 weeks after the second boost, monitored, and sacrificed on days 2 and 4 after infection as described above.

All in vivo experiments described were approved by the Institutional Animal Care and Use Committee at the University of North Carolina at Chapel Hill (animal protocol number: 21-056) in agreement with guidelines outlined by the Association for the Assessment and Accreditation of Laboratory Animal Care and the U.S. Department of Agriculture. Experiments were conducted under proper biosafety level 3 conditions, with personnel wearing full-body personal protective equipment (PPE) and HEPA-filtered respiratory protection.

### Plaque assay

To determine viral load in lungs the inferior lobe was harvested into vials containing 1 mL of PBS and glass beads. Samples were stored at -80 °C until further processing. Samples were thawed, homogenized, and serially diluted. Tissue supernatant dilutions were added to a monolayer of Vero E6 cells, overlayed with 0.8% agarose and incubated for two (SARS-CoV MA15) or three (SARS-CoV-2 MA10) days. Plaques were visualized using neutral red dye.

### Statistical modeling and analysis

Statistical testing, except mixed models, was performed in GraphPad version 10.0.1, GraphPad Software, Boston, Massachusetts USA.

Biological replicates, statistical tests, and post-tests are described in the figure legends. Mixed model performance for all three in vivo experiments was analyzed (Supplementary Fig. 7) using JMP® Pro version 17.1.0, JMP Statistical Discovery, Cary, North Carolina USA. Repeated measures per subject was used as a random variable.

### Reporting summary

Further information on research design is available in the Nature Portfolio Reporting Summary linked to this article.

## Data availability

Atomic coordinates and structure factors for the HexaPro-SS-Δstalk crystal structure generated in this study have been deposited in the Protein Data Bank (PDB) under accession code 8U1G. A reporting summary for this article is available as a Supplementary Information file. Source data are provided with this paper.

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

## Acknowledgements

We thank Dr. Kaci Erwin for technical assistance with mammalian cell culture. We acknowledge The University of Texas College of Natural Sciences and award RR160023 of the Cancer Prevention and Research Institute of Texas for support of the EM facility at The University of Texas at Austin. This work was funded in part by the Bill & Melinda Gates Foundation Investment ID INV-031624 (K.C., R.S.B., J.S.M.) and the National Institutes of Health research grant AI167966 (R.S.B.). Argonne is operated by UChicago Argonne, LLC, for the US Department of Energy (DOE), Office of Biological and Environmental Research under Contract DE-AC02-06CH11357.

## Author contributions

Conceptualization, C-L.H. and J.S.M. Investigation, C-L.H., S.R.L., E.H.M, L.Z, J.M.P., A.L.T., A.W., A.W., M.R.Z. and J.C.S. Visualization, C-L.H., S.R.L. E.H.M. and L.Z.; Writing – Original Draft, C-L.H., S.R.L. and E.H.M.; Writing – Reviewing & Editing, all authors. Supervision and funding, K.C., R.S.B. and J.S.M.

## Competing interests

C-L.H. and J.S.M. are inventors on U.S. patent application no. 63/188,813 ("stabilized S2 beta-coronavirus antigens"). S.R.L. and R.S.B. are inventors on U.S. patent application no. 11,225,508 ("Mouse-adapted SARS-CoV-2 viruses and methods of use thereof"). R.S.B. serves on the Scientific Advisory Board of Takeda, VaxArt, and Invivyd and has collaborations with Janssen Pharmaceuticals, Gilead, Chimerix, and Pardes Biosciences. The remaining authors declare no competing interests.
