## [Peer Review File · Nature Communications]

Prefusion-stabilized SARS-CoV-2 S2-only antigen provides protection against SARS-CoV-2 challengeReviewers' Comments:

Reviewer #1:

Remarks to the Author:

In this work, the authors describe a strategy to lock the sars cov 2 Spike S2 subunit in the prefusion state. The authors also tested vaccine candidates that contained tandem stem helix structures from different betacoronaviruses to induce stem helix specific antibodies. The modifications used to achieve this are guided by structure and described in detail. Vaccination with this vaccine results in broadly protective vaccine responses that protect against Sarscov2 and sarscov1 infections in a mouse model, and sera are cross-reactive among sarbecoviruses. The findings from these data are highly relevant and important for public health guidance and the quest for a universal pan-sarbeco vaccine.

The manuscript is very well written, with very clear figures. The data support the conclusions.

The manuscript is very complete as it is now, with much detail provided. It would be informative to map differences in antibody binding of polyclonal sera by crystallography after vaccination with the different constructs, similar to the work done by Andrew Ward at Scripps, although this might be beyond the scope of this story but might be addressed more in depth in the discussion section.

Reviewer #2:

Remarks to the Author:

This study by Hsieh, Leist et al., describes the use of a structure-guided engineering approach to generate a prefusion-stabilized stem S2 subunit antigen based on the previously described HexaPro S protein. The authors generated S2 variants with a stabilised stem (HexaPro-SS), the addition of stalk regions from other betacoronaviruses (HexaPro-SS-PentaStalks) or the deletion of the stalk region (HexaPro-SS- Δ stalk). X-ray crystallography and cryoEM on Hexapro-SS- Δ stalk revealed that the modified S2 did indeed adopt a prefusion conformation and illustrated antigen "breathing", which has been described for class-I fusogens. In murine immunization studies, the authors show that there is improved pseudovirus neutralization for sarbecoviruses (SARS-CoV-2 WT, Omicron BA.1, SHC014, SARS-CoV-1) and MERS when using sera from mice immunized with multiple antigens compared to single antigen immunization. Finally, they show that immunization with Hexapro S2 subunit antigens provide some protection against lethal mouse-adapted SARS-CoV-2 and SARS-CoV-1 challenge. In particular, Hexapro-SS shows protection comparable to that induced by immunization with full-length S, while HexaPro-SS- Δ stalk elicited inferior protection, suggesting a role for the stem helix in protection. The study is well executed and provides important insights into targeting the conserved epitopes in the S2 subunit through vaccination, potentially paving the way for next generation pan-coronavirus therapeutics and vaccines.

Major points:

1. In the prime-boost-boost studies (Figure S4), why did the study only extend to D4 (vs D7 in the other animal experiments)? It very hard to conclude that these animals survived when the study does not extend beyond the day that most of the control animals succumbed; and also quite challenging to compare to the other experiments. This should be updated or the interpretation should be rephrased. It cannot be concluded that all the animals surviving at D4 would actually have survived this challenge had the experiment been continued – the authors' own data in the other challenge studies shows animals succumbing at D5/6/7. Related to this point - given the limited protection seen in the SARS-CoV prime-boost immunization studies, would it be valuable to move the prime-boost-boost data (especially if updated to have the same time course as the other studies) to the main text?
2. In the Discussion, the authors note in lines 287-288: "Similarly, antibody 3A3, broadly reacting to MERS-CoV, SARS-CoV and 287 SARS-CoV-2 S, bound to a highly conserved region on the S2 apex50. Notably, these S2 antibodies preferentially bind to a conformation of the spike when the S2 apex is

open41, 49, 50.” It would have been valuable for the authors to characterize the antigenicity of their new constructs with known S2 mAbs. For example, do they have similar binding profiles for S2-targeting nAbs as HexaPro, or are epitopes differentially presented?

3. A major finding of this study is that an immune response devoid of RBD/NTD neutralizing antibodies (i.e. S2-only responses) can be protective against death in mouse models of SARS-CoV-2 and SARS-CoV infection. The authors clearly demonstrate in vitro neutralization against betacoronaviruses by the S2-only immune response. However, the reader is somewhat left to wonder whether protection against death is conferred by this S2-only neutralizing response, and/or the effector functions of neutralizing and non-neutralizing antibodies elicited. Have the authors characterized any aspects of the effector function responses and whether there are differences in the S2-only immunized groups compared to what is elicited by HexaPro? If such data is not readily available, the Discussion should address this limitation of the study in understanding the mechanistic basis of the protection observed.

Minor points:

1. Although it has previously been shown in the literature, it could be useful for less familiar readers if the sequence conservation in the distal S2 apex vs stalk among coronaviruses could be portrayed in the manuscript.
2. The experimental design for the animal studies needs to be more clearly described in the text. For example, the authors have not specified which timepoint the sera in the neutralization assays (lines 130-145) are collected at. Based on the immunization schema, it seems like it would be week 7, but this should be clearly stated.
3. A description (and/or citation) for the congestion scores should be provided.
4. The description of titers in lines 199-201 is hard to follow and could be rephrased.
5. Lines 233-235: “However, as previously observed in the single boost study, all but the HexaPro-SS immunization strategy conferred complete protection from mortality”. It would be important to state which single boost study is being referred to here: SARS-CoV-2 or SARS-CoV challenge?
6. Lines 249-250: “Collectively, these data show that only a double boost with HexaPro provides increased protection compared to a single boost (Fig. 5 f-j).” It would be important to clarify what “protection” in this sentence corresponds to: protection from infection? protection from clinical disease? protection from death? Also, it seems like the figure call out here is inaccurate as Fig. 5 does not have f-j panels.
7. The mouse-adapted virus used in these studies was adapted in 2020 from COVID-19 strains present at that time. With higher neutralization titers against Wuhan shown in Fig 4, protection vs mouse-adapted SARS-CoV-2 MA10 may be expected. Given the limited neutralization of Omicron BA.1 by the S2 constructs, it would be important to discuss what the implications of this would be regarding currently circulating SARS-CoV-2 variants.

Reviewer #1 (Remarks to the Author):

In this work, the authors describe a strategy to lock the sars cov 2 Spike S2 subunit in the prefusion state. The authors also tested vaccine candidates that contained tandem stem helix structures from different betacoronaviruses to induce stem helix specific antibodies. The modifications used to achieve this are guided by structure and described in detail. Vaccination with this vaccine results in broadly protective vaccine responses that protect against Sarscov2 and sarscov1 infections in a mouse model, and sera are cross-reactive among sarbecoviruses. The findings from these data are highly relevant and important for public health guidance and the quest for a universal pan-sarbeco vaccine.

The manuscript is very well written, with very clear figures. The data support the conclusions.

The manuscript is very complete as it is now, with much detail provided. It would be informative to map differences in antibody binding of polyclonal sera by crystallography after vaccination with the different constructs, similar to the work done by Andrew Ward at Scripps, although this might be beyond the scope of this story but might be addressed more in depth in the discussion section.

Answer: We thank the reviewer for their favorable comments and constructive suggestions. We have not yet been able to get the EMPEM technique from the Ward lab working, but we are keen to do so. In lieu of having these results available for the manuscript, we now discuss the potential benefit of structural characterization of polyclonal sera from S2-immunized mice in lines 343–346 and cite the literature describing the EMPEM technique by which future studies will be conducted:

“Thus, it will be interesting to structurally characterize the polyclonal antibody responses from the S2-immunized mice via EM^{66, 67} and investigate the epitopes targeted by the elicited antibodies. Functional epitopes that are recognized by neutralizing antibodies and antibodies associated with potent effector functions will inform additional antigen design efforts.”

Reviewer #2 (Remarks to the Author):

This study by Hsieh, Leist et al., describes the use of a structure-guided engineering approach to generate a prefusion-stabilized stem S2 subunit antigen based on the previously described HexaPro S protein. The authors generated S2 variants with a stabilised stem (HexaPro-SS), the addition of stalk regions from other betacoronaviruses (HexaPro-SS-PentaStalks) or the deletion of the stalk region (HexaPro-SS- Δ stalk). X-ray crystallography and cryoEM on Hexapro-SS- Δ stalk revealed that the modified S2 did indeed adopt a prefusion conformation and illustrated antigen “breathing”, which has been described for class-I fusogens. In murine immunization studies, the authors show that there is improved pseudovirus neutralization for sarbecoviruses (SARS-CoV-2 WT, Omicron BA.1, SHC014, SARS-CoV-1) and MERS when using sera from mice immunized with multiple antigens compared to single antigen immunization. Finally, they show that immunization with Hexapro S2 subunit antigens provide some protection against lethal mouse-adapted SARS-CoV-2 and SARS-CoV-1 challenge. In particular, Hexapro-SS shows protection comparable to that induced by immunization with full-length S, while HexaPro-SS- Δ stalk elicited inferior protection, suggesting a role for the stem helix in protection. The study is well executed and provides important

insights into targeting the conserved epitopes in the S2 subunit through vaccination, potentially paving the way for next generation pan-coronavirus therapeutics and vaccines.

Major points:

1. In the prime-boost-boost studies (Figure S4), why did the study only extend to D4 (vs D7 in the other animal experiments)? It very hard to conclude that these animals survived when the study does not extend beyond the day that most of the control animals succumbed; and also quite challenging to compare to the other experiments. This should be updated or the interpretation should be rephrased. It cannot be concluded that all the animals surviving at D4 would actually have survived this challenge had the experiment been continued – the authors' own data in the other challenge studies shows animals succumbing at D5/6/7. Related to this point - given the limited protection seen in the SARS-CoV prime-boost immunization studies, would it be valuable to move the prime-boost-boost data (especially if updated to have the same time course as the other studies) to the main text?

Answer: We thank the reviewers for their constructive comment. The main purpose of the double boost experiment was to increase beneficial effects on cross-protection by altering the immunization strategy. We showed before (Fig.6) that immunization with HexaPro as well as HexaPro-SS (same antigen for prime and boost) led to increased survivability of mice after SARS-CoV infection compared to non-immunized controls; however, changes in body weight were not affected. To reduce morbidity and mortality in our experimental cohort we decided to end the experiment on day 4 instead of day 7 to keep the number of animals that reach experimental endpoints for humane euthanasia as low as possible. We agree with the reviewers that the statement in the main text concerning “complete protection from mortality” has to be toned down (see lines **242–244** – now “conferred protection from mortality until the end of the study at day 4”).

2. In the Discussion, the authors note in lines 287-288: “Similarly, antibody 3A3, broadly reacting to MERS-CoV, SARS-CoV and 287 SARS-CoV-2 S, bound to a highly conserved region on the S2 apex50. Notably, these S2 antibodies preferentially bind to a conformation of the spike when the S2 apex is open^{41, 49, 50}.” It would have been valuable for the authors to characterize the antigenicity of their new constructs with known S2 mAbs. For example, do they have similar binding profiles for S2-targeting nAbs as HexaPro, or are epitopes differentially presented?

Answer: We thank the reviewer for this suggestion. We now include the binding data of S2-specific antibodies such as stem-helix-directed antibodies and fusion-peptide-directed antibodies in **Supplementary Fig. 3** and describe the results in lines **102–111**:

“To confirm that our engineering did not alter the antigenic surface of S2, we examined the binding of various S2-specific antibodies to the antigens (**Supplementary Fig. 3**). As expected, RBD-targeting antibody N3-1³³ did not bind to HexaPro-SS and HexaPro-SS- Δ stalk. Previous studies have reported that stem-helix-specific S2 antibodies show broadly neutralizing capability. Importantly, stem-helix antibodies IgG22²¹, S2P6¹⁸ and CC40.8²⁰ bound to HexaPro-SS at similar magnitudes as those observed for binding to the full S ectodomain S-2P and HexaPro, but not HexaPro-SS- Δ stalk, which lacks the epitope. Interestingly, fusion-peptide-directed antibodies CoV44-79²³ and CoV91-27 bound strongly to HexaPro-SS and weakly to the

full-length spikes (S-2P and HexaPro), suggesting that the fusion peptide is more accessible in the context of S2-only antigens.”

3. A major finding of this study is that an immune response devoid of RBD/NTD neutralizing antibodies (i.e. S2-only responses) can be protective against death in mouse models of SARS-CoV-2 and SARS-CoV infection. The authors clearly demonstrate *in vitro* neutralization against betacoronaviruses by the S2-only immune response. However, the reader is somewhat left to wonder whether protection against death is conferred by this S2-only neutralizing response, and/or the effector functions of neutralizing and non-neutralizing antibodies elicited. Have the authors characterized any aspects of the effector function responses and whether there are differences in the S2-only immunized groups compared to what is elicited by HexaPro? If such data is not readily available, the Discussion should address this limitation of the study in understanding the mechanistic basis of the protection observed.

Answer: Thank you for the constructive advice. We plan to fully characterize the nature of the protective responses in a follow-up manuscript. For now, we have included the limitation of this study and discussed the potential mechanisms of protection conferred by S2-only antigens in lines **335–342**:

A limitation of this study is that we did not assess the effector functions of the elicited antibodies⁵⁸, which in combination with antibody neutralization likely contributes to the observed protection. Antibodies that target membrane proximal regions of the related fusion proteins from influenza and HIV-1 appear to induce stronger antibody-dependent cellular cytotoxicity and complement deposition^{60, 61, 62, 63, 64}, which could help ameliorate disease. Cross-reactive S2-specific antibodies were also found to be more abundant in survivors of COVID-19 compared to non-survivors⁶⁵. It has been recently shown that non-neutralizing antibodies targeting HR1, fusion peptide, and HR2 can stimulate FcγR4 effector functions and convey protection to diverse sarbecoviruses²².

Minor points:

1. Although it has previously been shown in the literature, it could be useful for less familiar readers if the sequence conservation in the distal S2 apex vs stalk among coronaviruses could be portrayed in the manuscript.

Answer: We thank the reviewer for pointing this out. We have added **Supplementary Fig. 7** to show sequence conservation of the SARS-CoV-2 spike S2 subunit using the ConSurf server. Overall, the spike S2 subunit is highly conserved among 95 different coronavirus spike proteins.

2. The experimental design for the animal studies needs to be more clearly described in the text. For example, the authors have not specified which timepoint the sera in the neutralization assays (lines 130-145) are collected at. Based on the immunization schema, it seems like it would be week 7, but this should be clearly stated.

Answer: We thank the reviewers for pointing out the need for clarification. We added a sentence to the manuscript, lines **140-141**:

“Serum was collected one week prior to boost (week 2) and infection (week7), respectively (Supplementary Fig. 4). We then compared the ability of post-boost sera...”

3. A description (and/or citation) for the congestion scores should be provided.

Answer: We thank the reviewers for their comment. We added wording in the main text to clarify how congestion scores are defined, lines 169-171:

“(congestion score: 0–4; 0 = healthy pink lung, 1 = 25%, 2 = 50%, 3 = 75%, 4 = 100% of whole lung tissue exhibits dark red discoloration)”. The sentence before describes the method: “Gross evaluation of macroscopic changes in coloration of lung tissue at the time of sample collection”.

4. The description of titers in lines 199-201 is hard to follow and could be rephrased.

Answer: Thank you for your comment – we rephrased the passage accordingly:

Original:

Viral titers in the upper respiratory tract (**Fig. 6d**) were similar across all groups at around 10^5 PFU on day 2 after infection and 10^4 PFU on day 4 after infection. Only the two surviving mice in the HexaPro-SS-PentaStalks immunized group exhibited slightly elevated but not significantly different titers averaging 10^5 PFU. All groups showed similarly high viral titers in the lower respiratory tract at 10^7 PFU by day 2 after infection. Those titers decreased by 2 logs (10^5 PFU) for all groups except the HexaPro-SS-immunized group which exhibited more reduced lung titers at 10^4 PFU (**Fig. 6e**).

Updated (lines 204-210):

“Viral titers in the upper respiratory tract (**Fig. 6d**) were similar across all groups at around 10^5 PFU on day 2 after infection and 10^4 PFU on day 4 after infection. All groups showed similarly high viral titers in the lower respiratory tract at 10^7 PFU by day 2 after infection which decreased to 10^4 by day 4. The only exception were the two surviving mice in the HexaPro-SS-PentaStalks immunized group exhibiting slightly elevated titers (10^5 PFU) in the upper respiratory tract on day 4 as well as the HexaPro-SS-immunized group showing reduced lung titers (10^4 PFU) on day 4 after infection (**Fig. 6e**).”

5. Lines 233-235: “However, as previously observed in the single boost study, all but the HexaPro-SS immunization strategy conferred complete protection from mortality”. It would be important to state which single boost study is being referred to here: SARS-CoV-2 or SARS-CoV challenge?

Answer: We apologize for the lack of clarity and have thus reworded as follows:

Lines 236-244: “To investigate whether this immunization strategy can protect the mice from SARS-CoV, double-boosted mice were infected 8 weeks after the second boost (week 18) with 10^4 PFU mouse-adapted SARS-CoV MA15 and monitored daily for clinical signs of disease. Dedicated groups of mice for each immunization regimen were sacrificed for tissue collection on days 2 and 4 after infection (**Supplementary Fig. 5a**). Only triple HexaPro-immunized mice showed significant protection from weight loss whereas all other groups followed the control group trajectory (**Supplementary Fig. 5b**). However, all but the HexaPro-primed, HexaPro-SS-boosted immunization strategy conferred protection from mortality until the end of the study at day 4 (**Supplementary Fig. 5c**).”

6. Lines 249-250: “Collectively, these data show that only a double boost with HexaPro provides increased protection compared to a single boost (Fig. 5 f-j).” It would be important to clarify what “protection” in this sentence corresponds to: protection from infection? protection from clinical disease? protection from death? Also, it seems like the figure call out here is inaccurate as Fig. 5 does not have f-j panels.

Answer: We thank the reviewer for the comment. The sentence has been re-phrased and the correct figures are called out now: Lines 257-259:

“Collectively, these data show that only a double boost with HexaPro provides increased protection not only from mortality but also from changes in body weight after SARS-CoV infection compared to a single boost (Fig. 5 and Supplementary Fig.5).”

7. The mouse-adapted virus used in these studies was adapted in 2020 from COVID-19 strains present at that time. With higher neutralization titers against Wuhan shown in Fig 4, protection vs mouse-adapted SARS-CoV-2 MA10 may be expected. Given the limited neutralization of Omicron BA.1 by the S2 constructs, it would be important to discuss what the implications of this would be regarding currently circulating SARS-CoV-2 variants.

Answer: We thank the reviewers for pointing this important implication out. As the spike protein displays the most variation between the SARS-CoV-2 variants circulating in humans we think that updating our current constructs to better reflect circulating variants would confer increased protection. However, we feel that it is beyond the scope of the current manuscript to include updated constructs as well as utilizing mouse-adapted versions of different SARS-CoV-2 variants. Furthermore, the overarching goal is to find constructs that confer cross protection not only to various SARS-CoV-2 variants but also against different sarbecoviruses. We added a sentence to the Discussion to address this point:

Lines 346-348: “For future investigation, updating the constructs to currently circulating SARS-CoV-2 variants, such as BA.5, XBB1.5 and BA.2.86., may confer increased protection.”

Reviewers' Comments:

Reviewer #2:

Remarks to the Author:

We thank the authors for updating the manuscript to most of the concerns raised. One major concern remains.

The manuscript leans heavily on the finding that immunizations with prefusion-stabilized S2 constructs are able to protect mice from lethal SARS-CoV-2 and SARS-CoV challenges (lines 9 – 11; 47-48; 264-266, 285-287). Despite the changes in language introduced, this Reviewer still remains skeptical that the conclusion of protection against weight loss and lethality in SARS-CoV challenge for stabilized S2 constructs can be drawn from the data as shown. Figures 6 and S5 show similar weight loss profiles and survival, although this interpretation is made more challenging by the difference in time course of the two studies, and as previously raised, it cannot be concluded that animals surviving at D4 would survive the study if continued to D7. To allow for a more fulsome comparison and interpretation of the results, this Reviewer strongly believes that it would be valuable for the authors to display this weight loss and survival data together or side-by-side in the main text with the scale controlled. In addition, the authors state in their rebuttal that: "To reduce morbidity and mortality in our experimental cohort we decided to end the experiment on day 4 instead of day 7 to keep the number of animals that reach experimental endpoints for humane euthanasia as low as possible." It would be important for the authors to explicitly define in the text and figure legends the criteria for experimental endpoints for humane euthanasia. Indeed, to this Reviewer, it is not immediately obvious how the raw data in the "double boost SARS-CoV MA15" study in terms of weight loss (and corresponding experimental endpoints for humane euthanasia) and DIC is being portrayed on the survival plot. We believe that clarifications in this regard will be critical for the reader to be able to evaluate the outcome of this experiment and the conclusions derived.

Minor concerns

- Line 196: mortality rates of 10% are not possible in a group of only 5 mice.
- What dose of SARS-CoV-1 was used in the challenge studies? The methods and parts of the text say 10^4 (Lines 239, 469, 480) but in other areas it says 10^5 (lines 160, 192, 327).
- What was the ethical endpoint for the animal studies? The dotted line on the figures (5, 6, S5) seem to indicate that reaching 80% of starting body weight would be the endpoint, but the fact that some animals exceed this (see error bars) suggests that the endpoint may be 70%? Please specify.
- Line 343: COVID-19 typographical error
- In new supplementary figure 7, it would be useful to add information on which coronavirus strains were used in the analysis.

REVIEWER COMMENTS

Reviewer #2 (Remarks to the Author):

We thank the authors for updating the manuscript to most of the concerns raised. One major concern remains.

The manuscript leans heavily on the finding that immunizations with prefusion-stabilized S2 constructs are able to protect mice from lethal SARS-CoV-2 and SARS-CoV challenges (lines 9 – 11; 47-48; 264-266, 285-287). Despite the changes in language introduced, this Reviewer still remains skeptical that the conclusion of protection against weight loss and lethality in SARS-CoV challenge for stabilized S2 constructs can be drawn from the data as shown. Figures 6 and S5 show similar weight loss profiles and survival, although this interpretation is made more challenging by the difference in time course of the two studies, and as previously raised, it cannot be concluded that animals surviving at D4 would survive the study if continued to D7. To allow for a more fulsome comparison and interpretation of the results, this Reviewer strongly believes that it would be valuable for the authors to display this weight loss and survival data together or side-by-side in the main text with the scale controlled. In addition, the authors state in their rebuttal that: “To reduce morbidity and mortality in our experimental cohort we decided to end the experiment on day 4 instead of day 7 to keep the number of animals that reach experimental endpoints for humane euthanasia as low as possible.” It would be important for the authors to explicitly define in the text and figure legends the criteria for experimental endpoints for humane euthanasia.

Response: We thank the reviewer for the insightful comments. As requested, we have moved Supplementary Fig. 5 to Main Figure 7 to allow for a straightforward, side-by-side comparison of the two experiments.

We now explicitly define the euthanasia criteria and monitoring after the first description of the challenge experiments on lines 160-164 “...monitored daily for changes in body weight and signs of morbidity. As soon as animals reach 80% of their starting weight, they are subject to more stringent observation with weights measured twice a day as well as a visual check-in between measurements. Animals that approach 70% of their starting weight are humanely euthanized.”

We also added the following statement regarding our decision to end the double-boost experiment on day 4 to the main text, (lines 243-245): “To reduce morbidity and mortality in our experimental cohort, we decided to end the experiment on day 4 instead of day 7 to keep the number of animals that reach experimental endpoints for humane euthanasia as low as possible.”

Note that the SARS-CoV challenge experiment shown in Figure 6 demonstrates that the S2 constructs partially protect the mice from a lethal challenge with mouse-adapted SARS-CoV MA15 through day 7. Therefore, statements on lines 9-11 and 47-48 are supported by the data.

The experiment previously shown in Fig S5 (and now Fig 7) simply investigates whether immunizing first with full ectodomain enhances the response observed following boosts with the stabilized S2 constructs, but this was not observed, as we note “Only triple HexaPro-immunized mice showed significant protection from weight loss whereas all other groups followed the control group trajectory.” The main conclusion we draw from this experiment is stated as “Collectively, these data show that only a double boost with HexaPro provides increased protection not only from mortality but also from changes in body weight after SARS-CoV infection compared to a single boost.” In the following sentence, we note “Vaccination approaches with HexaPro in combination with two boosts of each construct did not protect from clinical signs of disease.”

We have deleted the sentence comprising previous lines 264-266 (“However, similar to the vaccination strategies involving a single immunogen with only one boost, the approach described here was largely able to protect mice from death in a lethal challenge with mouse-adapted SARS-CoV MA15”) given that the experiment ended on day 4.

The statement on previous lines 285-287 (“Notably, HexaPro-SS, even lacking the immunodominant S1 subunit, protected mice from SARS-CoV-2 lethal challenge and partially protected mice from SARS-CoV challenge, comparable to full-length S ectodomain”) is in reference to the data shown in Figures 5 and 6, and does not depend on the data shown in previous Supplementary Fig. 5 (now Figure 7). To make this clear, we have cited Figure 5 and Figure 6 appropriately in that sentence.

Indeed, to this Reviewer, it is not immediately obvious how the raw data in the “double boost SARS-CoV MA15” study in terms of weight loss (and corresponding experimental endpoints for humane euthanasia) and DIC is being portrayed on the survival plot. We believe that clarifications in this regard will be critical for the reader to be able to evaluate the outcome of this experiment and the conclusions derived.

Response: We double-checked every data point in the Source Data file and the figures showing the animal experiments. We found two single-data-point plotting errors, one in previous Fig S5c and one in previous Fig S5d (now Figure 7):

-Survival: For the HexaPro-SS-SS regimen, only 40% of animals survived, but the graph showed 80%. We exchanged panel C in Figure 7 with the updated data point.

-Congestion score day 2: HexaPro-HexaPro-HexaPro was displayed as 5 x 0, but there was one mouse with a 0.5 score. We exchanged panel D in Figure 7 with the updated data point.

No changes to the main text were required, as we did not call out those specific stats. The conclusions also remain unchanged.

Minor concerns

-Line 196: mortality rates of 10% are not possible in a group of only 5 mice.

Response: We thank the reviewer for catching this mistake. We have changed the sentence to “Among the groups of mice immunized with the different spike constructs, HexaPro, HexaPro-SS, and HexaPro-S2 showed similar protection with mortality rates of 20% (Fig. 6b).”

-What dose of SARS-CoV-1 was used in the challenge studies? The methods and parts of the text say 10^4 (Lines 239, 469, 480) but in other areas it says 10^5 (lines 160, 192, 327).

Response: We thank the reviewer for noting the discrepancy. We have made the following changes:

Lines 159-161: “Five weeks after boost, animals were infected with either 10^4 PFU SARS-CoV-2 MA10 or 10^4 PFU of the heterologous SARS-CoV MA15^{34, 35, 36} and monitored daily for clinical signs of disease...”

Lines 192-194: “In contrast to the SARS-CoV-2 MA10 challenge, none of the constructs provided significant protection from weight loss after challenge with 10^4 PFU of mouse-adapted SARS-CoV MA15 (Fig. 6a).”

-What was the ethical endpoint for the animal studies? The dotted line on the figures (5, 6, S5) seem to indicate that reaching 80% of starting body weight would be the endpoint, but the fact that some animals exceed this (see error bars) suggests that the endpoint may be 70%? Please specify.

Response: Correct, the endpoint was 70%, as noted in the response above and now in the text on lines 160-164.

-Line 343: COVID-19 typographical error

Response: We have corrected the typo.

-In new supplementary figure 7, it would be useful to add information on which coronavirus strains were used in the analysis.

Response: We have now included the alignment file as a supplementary file for the manuscript.